# Evaluating Generalization Capabilities of LLM-Based Agents in Mixed-Motive Scenarios Using Concordia

Chandler Smith[1,2]    Marwa Abdulhai[3]    Manfred Diaz[4]    Marko Tesic[5]
Rakshit S. Trivedi[6]    Alexander Sasha Vezhnevets[7]    Lewis Hammond[1,2]    Jesse Clifton[1,8]
Minsuk Chang[7]    Edgar A. Duéñez-Guzmán[7]    John P. Agapiou[7]    Jayd Matyas[7]
Danny Karmon[9]    Dylan Hadfield-Menell[6]    Natasha Jaques[7,10]    Tim Baarslag[11,12]
Jose Hernandez-Orallo[13,5]    Joel Z. Leibo[7]

**Concordia Contest Participants with Notable Contributions**
Akash Kundu, Aliaksei Korshuk, Ananya Ananya, Arrasy Rahman, Avinaash Anand Kulandaivel, Bain McHale, Beining Zhang, Buyantuev Alexander, Carlos Saith Rodriguez Rojas, Caroline Wang Chetan Talele, Chenao Liu, Chichen Lin, Diana Riazi, Di Yang Shi, Emanuel Tewolde, Elizaveta Tennant, Fangwei Zhong, Fuyang Cui, Gang Zhao, Gema Parreño Piqueras, Hyeonggeun Yun, Ilya Makarov, Jiaxun Cui, Jebish Purbey, Jim Dilkes, Jord Nguyen, Lingyun Xiao, Luis Felipe Giraldo, Manuela Chacon-Chamorro, Manuel Sebastian Rios Beltran, Marta Emili García Segura, Mengmeng Wang, Mogtaba Alim, Nicanor Quijano, Nico Schiavone, Olivia Macmillan-Scott, Oswaldo Peña, Peter Stone, Ram Mohan Rao Kadiyala, Rolando Fernandez, Ruben Manrique, Sunjia Lu, Sheila A. McIlraith, Shamika Dhuri, Shuqing Shi, Siddhant Gupta, Sneheel Sarangi, Sriram Ganapathi Subramanian, Taehun Cha, Toryn Q. Klassen, Wenming Tu, Weijian Fan, Wu Ruiyang, Xue Feng, Yali Du, Yang Liu, Yiding Wang, Yipeng Kang, Yoonchang Sung, Yuxuan Chen, Zhaowei Zhang, Zhihan Wang, Zhiqiang Wu, Ziang Chen, Zilong Zheng, Zixia Jia, Ziyan Wang

[1]Cooperative AI Foundation    [2]University of Oxford    [3]UC Berkeley    [4]Quebec Artificial Intelligence Institute    [5]Leverhulme Centre for the Future of Intelligence, University of Cambridge    [6]MIT    [7]Google DeepMind    [8]Center on Long-Term Risk    [9]Google Research    [10]University of Washington    [11]Centrum Wiskunde & Informatica    [12]Utrecht University    [13]Universitat Politècnica de València

## Abstract

Large Language Model (LLM) agents have demonstrated impressive capabilities for social interaction and are increasingly being deployed in situations where they might engage with both human and artificial agents. These interactions represent a critical frontier for LLM-based agents, yet existing evaluation methods fail to measure how well these capabilities generalize to novel social situations. In this paper, we introduce a method for evaluating the ability of LLM-based agents to cooperate in zero-shot, mixed-motive environments using Concordia, a natural language multi-agent simulation environment. Our method measures general cooperative intelligence by testing an agent's ability to identify and exploit opportunities for mutual gain across diverse partners and contexts. We present empirical results from the NeurIPS 2024 Concordia Contest, where agents were evaluated on their ability to achieve mutual gains across a suite of diverse scenarios ranging from negotiation to collective action problems. Our findings reveal significant gaps between current agent capabilities and the robust generalization required for reliable cooperation, particularly in scenarios demanding persuasion and norm enforcement.

39th Conference on Neural Information Processing Systems (NeurIPS 2025) Track on Datasets and Benchmarks.

# 1 Introduction

Large Language Models (LLMs) have recently demonstrated impressive capabilities as the foundation for generative agents that can engage in naturalistic social interactions [70, 44, 63]. However, despite rapid advancements in the abilities of these LLM-based agents, research on their generalization capabilities in cooperative, mixed-motive scenarios has received less attention. This gap stands in stark contrast to supervised learning research, where evaluation methodologies explicitly prioritize generalization to new, previously unseen examples [69, 24, 16, 42]. Addressing this gap is essential to being able to train and deploy artificial agents that can coordinate and cooperate with both humans and other agents in dynamic environments. The assessment of cooperative generalization in LLM-based agents presents unique challenges. It requires carefully designed environments that balance structure with open-endedness, flexible mechanisms for communication and action, and nuanced evaluation metrics that capture the multi-faceted nature of cooperation. Current benchmarks typically evaluate performance in either fixed-partner settings or fully competitive games, failing to capture the critical middle ground where both cooperation and self-interest matter [8, 14, 65].

To address this gap, we present the 2024 NeurIPS Concordia Contest (hereafter, The Contest), a principled approach to evaluating generalization in LLM-based agents within mixed-motive social interactions [54]. Following in the tradition of the Melting Pot contest for multi-agent reinforcement learning [35, 2, 59], The Contest adopts the fundamental premise that the evaluation of cooperative agents should focus on generalization to novel social contexts—especially the ability to maintain cooperative behavior when interacting with unfamiliar social partners.

This paper makes the following key contributions:

1. We present a principled argument for measuring cooperative intelligence centered on generalization, drawing parallels with evaluation practices in supervised and reinforcement learning while highlighting the unique demands of multi-agent social interaction.

2. We introduce a first-of-its-kind evaluation framework consisting of five LLM-simulated environments that test distinct aspects of cooperation, including strategic communication, social coordination under uncertainty, negotiation, and collective action. Each environment incorporates the "veil of ignorance" principle to assess zero-shot generalization capabilities.

3. We provide detailed empirical evidence from the NeurIPS 2024 Concordia Contest, revealing the strengths and limitations of both current LLM-based agents and our proposed framework. Our analysis demonstrates that while some agents achieved moderate success in negotiation tasks, they struggled significantly with complex coordination scenarios requiring persuasion and norm enforcement. These findings highlight specific capability gaps that must be addressed to develop agents capable of robust cooperative generalization.

# 2 Related Work

**Evaluating Multi-Agent Systems.** Multi-agent systems (MAS) have historically been evaluated in either competitive or cooperative settings. Competitive benchmarks assess agents' abilities to defeat others in adversarial games such as chess [52], Go [53], poker [11], and StarCraft [65]. Cooperative benchmarks study coordination with teammates under full or partial observability [29, 6, 15, 45]. Mixed-motive interactions, where agents have overlapping but not identical goals, more realistically capture real-world social dynamics.. The Melting Pot benchmark introduced mixed-motive evaluation for reinforcement learning agents, testing generalization to novel co-players and scenarios [35]. Subsequent work has explored fine-grained metrics of social influence, inequity aversion, and partner adaptation in similar environments [27, 30, 3]. Our work extends this framework to LLM-based agents, evaluating their ability to generalize cooperative behavior through natural language in mixed-motive settings.

**LLM-Based Agents and Social Interaction.** LLMs have demonstrated emergent social capabilities in open-ended environments. Generative agents built from LLMs can model memory, intent, and behavior trajectories [44], while environments like Concordia formalize LLM interactions in mediated multi-agent games [63]. Interactive benchmarks such as Sotopia [70], ChatArena [67], and Light [61] explore social reasoning and alignment using structured games and narratives. Recent studies examine LLMs' theory of mind [60], moral reasoning [4, 1], and social preference modeling [10],

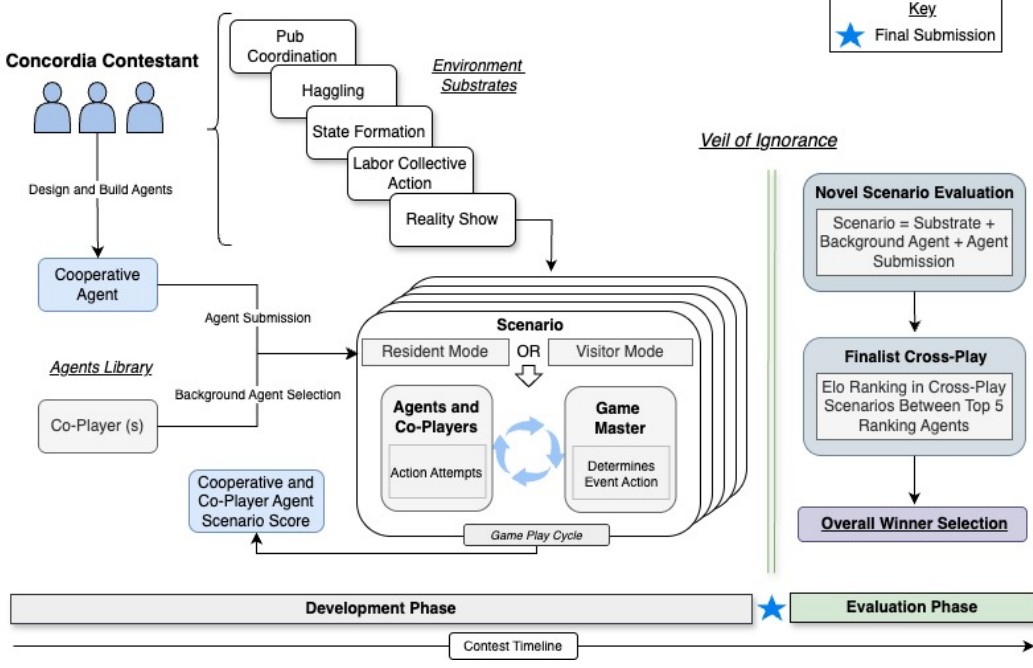

Figure 1: Overview of the 2024 NeurIPS Concordia Contest Framework. Contestants (top left) design and submit agents that, during the Development Phase, interact with background co-players across five cooperation-eliciting substrates (Pub Coordination, Haggling, State Formation, Labor Collective Action, Reality Show). Each scenario is run in either Resident and Visitor modes (see Section 4.3) under the orchestration of a Game Master, which mediates action attempts, determines resulting events, issues event statements and observations, and computes the agent's and co-player's scores. Marked by the blue star, contestants submit their final agents to the Evaluation Phase, which proceeds under a "veil of ignorance". Agents are first Elo-ranked in novel scenarios, then the top five performing agents engage in a cross-play round, which is used to determine the overall winner.

but typically test isolated skills in controlled settings. Other work explores multi-agent LLM coordination through debate [68], negotiation [37, 13], and voting-based deliberation [57, 32]. Further research examines complex multi-agent LLM interactions, including cooperative negotiations [41] and sophisticated strategic social reasoning [19]. Other authors have highlighted the potential risks from these interactions, including concerning escalation dynamics [47] and potential mis-coordination [22]. We build on these insights to design structured, generalizable benchmarks that evaluate whether LLM agents can maintain coherent cooperative strategies across a variety of social partners and scenarios.

**Cooperation in Agents.** Early approaches to measuring and training cooperative agents were focused on the emergence of cooperative behavior, norms, and general conventional patterns of cooperation from *tabula rasa* RL approaches [35, 64]. However, LLMs' increasing abilities to understand and interact with humans through natural language has shifted the field's interest towards producing agents that display cooperative intelligence when embedded in human contexts.

**Generalization in Cooperative Intelligence.** Generalization—the capacity to perform well in previously unseen settings—is a foundational problem in machine learning [62, 24]. In reinforcement learning, generalization challenges arise due to active data collection, sparse reward structures, and the tendency to overfit environment dynamics [16, 20]. Evaluation techniques such as domain randomization [58] and contextual MDPs [31] address this by diversifying environment conditions. The problem of multi-agent generalization is especially difficult due to the non-stationarity induced by co-adapting agents [26, 39, 17]. Evaluating an agent's cooperative intelligence requires assessing its behavior across diverse co-players, incentive structures, and social roles [18]. This includes testing whether agents can flexibly adapt to novel partners while avoiding brittle, overfit strategies. This area has gained traction in MARL settings, yet remains underexplored in LLMs [48]. Our evaluation methodology adapts these ideas to the LLM setting, using structured social tasks to test generalization over both environmental conditions and partner behaviors.

# 3 A Protocol for Measuring General Cooperation in Concordia

We take inspiration from the seminal work of Legg and Hutter [34], who provide a formal definition of intelligence based on the idea that "intelligence measures an agent's ability to achieve goals in a wide range of environments". In multi-agent settings, we can (in principle) separate the environment from the other strategic agents within it. Thus, an individual's cooperative intelligence must encompass not only its ability to achieve goals in a wide range of environments, but also when interacting with a wide range of co-players.

In particular, given our emphasis is on cooperation, we focus on an individual agent's (henceforth the *focal* agent) achievement of goals that represent mutual gains across different sets of co-players in different environments. We therefore restrict our attention to different 'cooperation-eliciting' settings, in which a focal agent performs well if (and only if) it can take advantage of opportunities for mutual gains whenever they arise. In the subsections below, we make these ideas more precise and introduce some of the key terminology used in the Concordia framework.

## 3.1 Generalization

To understand cooperative generalization in multi-agent settings, and inspired by previous work [35, 2], we separate scenarios into a *substrate*—the component capturing the rules and dynamics that govern the environment in which the agents interact—from the *background population* of agents embedded in it. This separation of concerns facilitates measuring the focal agent's performance across multiple interaction rules—as captured by a variety of substrates—and its robustness to interact with a wide-ranging set of co-players strategies—as determined by the structure of the population of interacting agents.

**Substrate-Aware Generalization.** To measure the focal agent's ability to generalize across varying scenarios, we require a collection of substrates $E = \langle E_1, E_2, \ldots, E_m \rangle$. Each substrate determines a set of rules and dynamics that govern the players' interactions. More precisely, we model a *substrate* as an $n$-player Partially-Observable Stochastic Game without Rewards (POSG\R), that we denote by $E = \langle \mathcal{S}, \mathcal{O}, \mathcal{A}, p \rangle$. The substrate consists of an underlying state space $\mathcal{S}$, a joint observation space $\mathcal{O} = O_1 \times O_2 \times \ldots O_n$, a joint action space $\mathcal{A} = A_1 \times A_2 \times \ldots \times A_n$, and a probabilistic function $p : \mathcal{S} \times \mathcal{A} \to \Delta(\mathcal{S} \times \mathcal{O})$ that determines the state transitions and the observations of each agent based on the previous state and the agents' joint action. The $i^{\text{th}}$ agent's policy $\pi_i$ is *substrate-admissible* for a substrate $E$ if it defines a distribution in $\Delta(A_i)$ (where $\Delta(X)$ denotes the set of probability distributions over set $X$) for any history of the agent's observations and actions $h \in H_i = (O_i \times A_i)^*$. Our focus on establishing the cooperative capabilities of the focal agent's strategy requires that the substrates in E capture interaction rules where there exist opportunities for cooperation among the players. We formalize this requirement later in Section 3.2.

**Population-Aware Generalization.** The second axis of generalization capabilities concerns the focal agent's ability to interact with diverse co-player populations. We introduce two measures to gauge such capacity. Our understanding of what co-player or population-aware generalization entails is motivated by the problem of robustness to mutations in evolutionary game theory [55, 56, 5]. Intuitively, we view the focal agent as successfully exhibiting cooperative intelligence if it follows patterns of cooperation that (i) resist the emergence of non-cooperative strategies, (ii) remain stable whenever possible, (iii) change only to adopt improved patterns of cooperation. To incorporate these desiderata when measuring an agent's population-aware generalization, we designed two settings that evaluate such capabilities.

- First, we consider **resident scenarios** $\mathcal{G}_R = \langle E, \rho, \texttt{resident} \rangle$ where a background strategy $\rho$ is specified. When evaluating a focal strategy $\pi$ in this scenario, we instantiate a population consisting of a majority of agents playing $\pi$ and a minority playing $\rho$. These resident scenarios have the capacity to measure the stability of the focal agent's cooperation with other cooperative strategies if $\rho$ is cooperative [desideratum (ii)] or, otherwise, its robustness to the invasion of non-cooperative strategies [desideratum (i)], both in cases where the focal strategy represents a majority of the population.

- Second, we examine **visitor scenarios** $\mathcal{G}_V = \langle E, \rho, \texttt{visitor} \rangle$, which invert this population structure: here we instantiate a majority of agents playing the background strategy $\rho$ and a minority agent playing the focal strategy $\pi$. If $\rho$ is cooperative, a visitor scenario measures

the focal agent's ability to identify and adapt its behavior to conform to social norms and conventional patterns of behavior [desideratum (iii)] whenever its strategy $\pi$, even though possibly cooperative, may represent non-conventional behavior adopted by only a minority of the population.

These *resident* and *visitor* configurations are special cases of generalized mixed population compositions. They represent controlled variations of the population structure that allow us to properly assign credit to the focal agent's cooperative capabilities.

## 3.2 Scenarios with Opportunities To Cooperate

To measure the cooperative capabilities of AI agents, we focus on substrate-aware and population-aware generalization in the context of cooperation-eliciting scenarios, which provide focal agents the opportunity to achieve their own goal only insofar as they can identify and achieve mutual gains. Accordingly, we make the following structural assumptions about the substrates, populations, and scenarios that result from their combination.

Drawing inspiration from Schelling's simplification of complex multiplayer interactions into *binary choices* [50], we design *cooperation-eliciting* scenarios by first (i) partitioning the players' collective strategy space into distinct sets of cooperative and non-cooperative strategies; and (ii) ensuring that they (the substrates) support a variety of such strategies, typical of those that might be seen in the context of human cooperation and competition. Previous research exploring cooperation among AI agents [36, 27, 40] has effectively utilized this approach to differentiate between human-like cooperative and non-cooperative behaviors in general-purpose multi-agent contexts.

For a background strategy $\rho$ and a focal strategy $\pi$, we say the combination is substrate-compatible with substrate $E$ if both $\rho$ and $\pi$ are admissible policies for the substrate. We then create a variety of background strategies $\rho$ based intuitively on common patterns of human behavior in mixed-motive settings (such as naive altruism, stubborn strategies, and conditional cooperation). These background strategies are designed such that the overall evaluation scenario is cooperation-eliciting. In these scenarios, we can leverage prior knowledge of what counts as cooperative behavior (allowing us to sidestep many well-known problems involving interpersonal comparisons of welfare [23, 51, 66, 25, 9]) to score the focal agent based on the proportion of agents in the instantiated population that play jointly-cooperative strategies. Given a scenario $G = \langle E, \rho, m \rangle$ (where $E$ is a substrate, $\rho$ is a background strategy, and $m \in \{\text{resident, visitor}\}$ is the mode) and a focal strategy $\pi$, we write the score as $s(G, \pi) \in \mathbb{R}$.

# 4 The 2024 NeurIPS Concordia Contest

The measure of cooperative intelligence introduced in Section 3 represents an ideal standard for eliciting such capabilities for AI agents. However, the operationalization of this measure in practice faces limitations on computational budget, time, and complexity restrictions over the design of substrates and populations of players. The Contest represented an instantiation of this protocol that leveraged principled design choices to balance practical restrictions with measuring LLM-based agents' cooperative capabilities. In this section, we offer a detailed description of the rationale behind our design decisions.

## 4.1 The Veil of Ignorance

The Contest evaluation protocol represented an elicitation mechanism that communicated to participants the designers' preferences for solutions with a set of desired characteristics. Similar to studies on generalization in supervised learning, The Contest specification compelled participants to develop agents with zero-shot intelligence, i.e. agents were designed behind a "veil of ignorance" [46]. Participants had to build an agent dropped into a novel multi-player scenario unaware of the specific context characteristics or the strategies of the other co-players they may interact with. Consequently, to succeed in these unfamiliar scenarios, an agent must demonstrate the ability to zero-shot generalize to variations of contexts and co-players.

**The Contest Structure.** As illustrated in Figure 1, The Contest was organized into two phases: a development phase, where participants submitted focal agents for evaluation against a set of publicly

available scenarios, receiving feedback on their performance, and an evaluation phase where the submitted agents were assessed on a set of held-out scenarios to gauge their zero-shot generalization capabilities. This separation, equivalent to the train-test divide in supervised learning, prevented participants from co-designing their solutions by exploiting particularities of the substrates or the behaviors of the populations of co-players.

Moreover, we communicated the protocol defined in Section 3 to participants. Therefore, knowledge of this evaluation metric requires participants to design focal agents that can identify *cooperation-eliciting substrates* and adapt to *cooperation-oriented populations* (Section 3.2). We underpinned this requirement by guaranteeing that both properties held for the scenarios in both phases, as explained in Section 4.3. Accordingly, an agent optimally designed to perform best in The Contest would need first to determine what constitutes cooperative behavior within an unknown context by gradually removing, through interactions, the "veil of ignorance" imposed by The Contest specification. This combination of design choices aims to stimulate the construction of agents that align with our definition of cooperative intelligence.

## 4.2   Focal Agent Design

We conceived a focal agent's policy as the composition $\pi \equiv \text{LLM}(f(o))$, which may involve one or more LLM API call functions $\text{LLM} : \mathcal{O} \to \Delta(\mathcal{O})$ and wraps the output of a scaffolding function $f : \mathcal{O} \to \mathcal{O}$, a concept initially proposed by the Concordia framework [54], the open-source platform for generative agent-based modeling that powered The Contest.[1]Scaffolding functions can be implemented as sections of agent code that enable sophisticated capabilities beyond prompt engineering, such as maintaining persistent memory, implementing numerical computations, enforcing logical constraints, and preprocessing complex observations. Participants were only required to submit scaffolding functions.

**Rationale.** The Contest and its participants gain significant advantages from this separation. Firstly, it facilitates the development of specialized agents by allowing the design of tailored cooperative scaffolds around the general-purpose capabilities of LLM, such as in-context learning. Secondly, it promotes fairness in the competition by levelling the playing field among contestants and preventing those with larger budgets from gaining an unfair advantage through access to more powerful LLMs, thereby also removing a confounder in performance comparisons. Lastly, it introduces a layer of flexibility for The Contest organizers to reduce the impact of LLM API resources on The Contest outcomes, especially when additional funding or computational resources are available. For The Contest, we selected the Gemma 2 model (9 billion parameters, instruction-tuned) as our LLM API provider, as it represented an effective balance between API call costs and overall model performance at the time of design [21]. Gemma-2 was selected after careful evaluation of other SOTA models at the time of contest inception, including Llama and Mistral models, and emerged as the most performant model per cost. The single model design choice serves critical purposes for fair evaluation: (1) it ensures a level playing field by preventing teams with larger computational budgets from gaining unfair advantages, (2) it provides consistency across all agent evaluations, eliminating model capability as a confounding variable, and (3) it makes The Contest accessible to broader participation.

## 4.3   Scenario Design

The Contest scenarios were derived from five text-based configurable substrates: REALITY SHOW, PUB COORDINATION, HAGGLING, LABOR COLLECTIVE ACTION, and STATE FORMATION problems that are structurally similar to (sequential) social dilemmas [36]. We present more details of these substrates in Table 1. We generate the substrates of the development and evaluation phases by generating variations of the time and place where these dilemmas occur and by varying the number of players that interact every time. The five substrates are *cooperation-eliciting*; thus, identifying players' cooperative and non-cooperative behavior is a built-in characteristic in their design.

Correspondingly, the populations in the development phase that parameterized these substrates consisted of *visitor* and *resident* populations of *cooperation-oriented* co-players, where The Contest designers developed the background players. However, during the evaluation phase, the scenarios were parameterized by *visitor* and *resident* populations containing pre-defined held-out background

---

[1]Designed with Concordia v1.8.9 at `https://github.com/google-deepmind/concordia`.

players created by The Contest designers, and the agents of the top-five best-performing submissions on the pre-defined scenarios.

Most combinations of substrates and background populations that composed The Contest scenarios were explicitly designed to be *cooperation-eliciting* and *cooperation-oriented*, respectively, as defined in the protocol Section 3.2. However, we did include non-cooperation-oriented populations of co-players in some evaluation-phase scenarios to verify whether the submitted agents were able to adapt to context and co-players that did not represent an opportunity to cooperate (desideratum (i), Section 3.1). The test scenarios contain multiple opportunities for exploitation of irrational behavior, encompassing vulnerability to exploitation by others and suboptimal decision-making. The protocol communicated to the participants was to design agents capable of identifying and executing cooperative strategies while minimizing irrational losses.

Table 1: Substrate Design. We also include tags that characterize each substrate.

| Substrate | Description |
| --- | --- |
| Reality Show | Contestants play a series of mini-games (Prisoner's Dilemma, Chicken, Stag Hunt) alternating between communication phases (strategy discussion) and action phases (committing to decisions). This substrate examines the emergence of cooperative norms, reputation effects, and strategic communication. **Tags**: discouraging antisocial behavior, persuasion, calculation, convention following |
| Pub Coordination | Players choose pubs to attend with friends—there is no explicit conflict but a tension between group coordination vs. individual preference. Unexpected closures introduce incomplete information and social scenes allow negotiation, persuasion, and information sharing. Success requires aligning choices, managing relationships, compromise, and adapting to change. **Tags**: coordination, persuasion, hidden information, social networks |
| Haggling | In Fruitville, 2–4 merchants negotiate prices over multiple bargaining rounds. This rewards mutually beneficial deals while balancing individual profit and long-term relationships. The payoff structure encourages agreements but includes tension between personal gain and partner willingness. **Tags**: negotiation, calculation |
| Labor Collective Action | Workers face a collective action problem: strike or continue working. Enough workers striking pressures the boss for higher wages, benefiting all, but individuals may defect to earn wages. Multi-day rounds allow strategy evolution, reputation effects, and peer influence. Characters include workers, a labor organizer, and a boss, with wage cuts and power dynamics shaping cooperation. **Tags**: discouraging antisocial behavior, persuasion, calculation |
| State Formation | Two villages threatened by raiders must negotiate an alliance. Village elders (main characters) and influential villagers (supporting) engage in diplomatic bargaining and public-goods provision. This tests an ability to satisfy multiple stakeholders, follow through on agreements, and manage constituency demands. **Tags**: negotiation, persuasion |

## 5 Evaluation Methodology and Results

This section presents our empirical evaluation pipeline and summarizes main findings from The Contest. A total of 197 participants contributed, with 25 teams submitting agents. Each agent was initially evaluated across a diverse set of cooperative substrates detailed in Table 1, after which the top six agents advanced to a tournament-style evaluation to determine the final cooperative rankings. To analyze performance, we collected and integrated four data sources: (1) agent rankings, (2) inferred ability profiles, (3) submitted policy code, and (4) participant-authored survey responses describing agent design and strategy.

**Agent Rankings.** The primary methodology we use to compare performance between agents is the Elo rating system (Table 3) due to its simplicity and widespread use in evaluating agent interactions [7, 35, 43, 28, 38]. We adopt Elo ratings rather than raw scores because each environment substrate has different randomized settings, making raw scores incomparable both between agents and across scenarios. While Elo rankings can be sensitive to noise and are not always stable, particularly when differences between agents are small, it remains a practical and interpretable metric for summarizing relative performance. In our contest, where many agents showed marginal gains over the baseline, Elo still helped surface meaningful distinctions in decision quality and strategic behavior. To alleviate known limitations of Elo, we also implemented a suite of evaluation metrics including Iterative Maximal Lotteries [32], Copeland, and Ranked Pairs (Tables 4 to 6). These methods allow us to explore how rankings might shift under different aggregation criteria and offer an interpretable view into how agents perform under different assumptions about fairness and transitivity. Across all ranking methods, the top three agents consistently emerged as the strongest performers, indicating robust cooperative behavior across evaluation criteria. All ranking data can be found in Section 9.4.1.

**Agent Scores.** Raw scores were first rescaled to the $[0, 1]$ interval with a per-scenario min–max normalization based on scenario theoretical minimum and maximum. Across the test bed, the average agent achieved a mean score of $0.426 \pm 0.005$ (SE), leaving considerable room for future progress (individual means in Figure 4). Performance was highly scenario-dependent (see Figure 3 left). In certain scenarios, agents approached the maximum achievable score, while their performance was notably lower in others.

During the design phase, each substrate was annotated with tags indicating the (cooperative) capabilities we hypothesized would be required for strong performance. Tags were 'pre-registered', i.e., tagging was completed during task design, before the competition began, and thus was not influenced by participants' submissions or their performance. For clarity, we provide an explanation of each tag in Table 13. We explore the effects of tags on score by fitting a hierarchical Beta-regression model for each scenario tag with agent-specific random slopes. The tag effects were given a joint multivariate-normal prior whose covariance followed an LKJ distribution accounting for correlations between tags (see Figure 5). We found that agents struggled in scenarios requiring persuasion, convention following, negotiation, discouraging antisocial behavior, and coordination, each reducing the average agent's expected performance by between ten and twenty percentage points (see Figure 3 right). Importantly, the majority of cooperative tags negatively affected agent scores, i.e. agents struggle more when a scenario requires these capabilities, which is the behavior we would expect if the benchmark is successfully probing the targeted cooperation capabilities.

Lastly, we generated a bayesian mixed-effects beta regression model, with a random intercept for each scenario and the rational agent as the baseline reference. This shows that the majority of agents either performed similarly to or significantly worse than the rational agent. Only five agents demonstrated significantly higher performance compared to the rational agent baseline (see Figure 2).

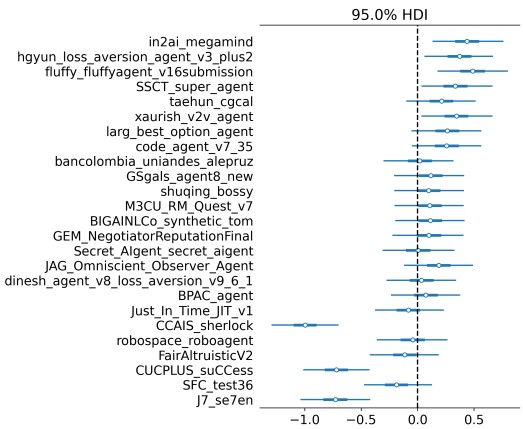

Figure 2: Posterior distributions (mean and 95% HDI) of agent performance (log-odds difference) relative to the rational agent baseline. Vertical dashed line indicates no difference.

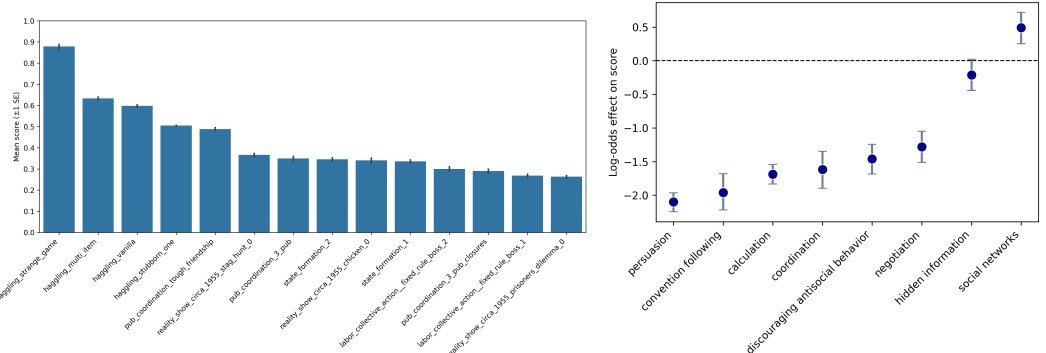

Figure 3: **Left:** Mean score for each scenario (error bars are ±1 SE). Agents performed relatively poorly in most scenarios, however in several scenarios their average score exceeded 50% of the theoretical maximum, and in one case approached 90%. **Right:** Posterior means and 94% highest-density intervals of the tag coefficients from the hierarchical beta-regression with LKJ-correlated priors. Nearly all cooperative tags have negative coefficients, meaning that the presence of these tags lowers agents' scores. This pattern indicates that agents struggled when scenarios required cooperation, as expected for a cooperation-eliciting benchmark.

**Agent Ability Profiles.** To further explain the patterns in agent performance beyond aggregate scores, we interpret scenario tags, substrates, and agent roles as demands. Combining these demands with the observed scores of each agent, we infer the latent capabilities that drive each agent's performance and explain whether agents systematically succeed or struggle under particular conditions. We use Measurement Layouts [12], hierarchical Bayesian models that infer each agent's latent capabilities from the observed demand-score structure. As shown in Figure 6, the persuasion capability is the main factor that distinguishes the top-performing agents from the rational-agent baseline. Conversely, a weak ability to discourage antisocial behavior characterizes the lowest-scoring agents. Finally, predictive modeling with Measurement Layouts, linear regression, and XGBoost using demands as input features significantly outperforms a constant baseline model (Figure 7), confirming that the tag, substrate and role annotations capture meaningful structure in agent performance. These results also suggest that the proposed substrates effectively differentiate agents based on their ability to generalize across different cooperative demands.

**Qualitative Analysis.** Our qualitative methodology consists of four complementary modules. First, we conduct a policy adherence assessment, in which an LLM is asked to summarize the intended strategy of each agent from the participant code and judge whether its observed actions in the agent log history are consistent with the strategy in the code. Second, we produce a behavioral summary asking the LLM to generate a concise 3–5 sentence description of the agent's negotiation style, highlighting attributes such as cooperativeness, adaptability, and concession pacing, from the agent's log history. Third, we administer a set of ten standardized Likert scale assessments covering traits such as fairness, empathy, aggressiveness, and consistency; an LLM assigns a 1 to 5 rating to each trait along with a brief rationale grounded in dialogue. These analyses allow us to understand further what makes agent policies successful beyond the Elo score ratings.

In Figure 8, several common failure modes emerged, including distraction from task objectives, selfish decision-making, and an inability to maintain goal alignment—often attributable to limitations in the underlying language model. These issues were evident in qualitative analyses, which revealed inconsistencies between intended and observed behavior, and were further supported by low Likert-scale ratings on traits such as fairness, empathy, and consistency. Despite these challenges, the Concordia framework enabled a clear and interpretable evaluation of generalization. Agents that performed well across diverse scenarios demonstrated greater consistency in cooperative strategy, both in behavioral summaries and quantitative scores. Likert evaluations correlated with higher returns, and qualitative examples confirmed that cooperative behavior was often aligned with stronger performance. These findings highlight the value of structured annotations and scenario diversity in diagnosing agent capabilities and evaluating generalization in multi-agent social environments.

# 6  Conclusion

The Concordia Contest represents a significant step toward rigorous evaluation of generalization in LLM-based agents within mixed-motive social contexts. Building on the Melting Pot competition, the contest offers a principled framework for assessing how well agents transfer social strategies to novel settings and unfamiliar partners. Unlike benchmarks focused on static task completion, Concordia centers on dynamic interaction, enabling fine-grained analysis of how cooperation emerges—or breaks down—under different conditions.

Our results highlight both the promise and current limitations of cooperative LLM-based agents. While some agents achieve strong performance in specific scenarios, few generalize effectively across structurally diverse and socially distinct environments. Performance disparities are often attributable to failures in goal tracking, susceptibility to selfish behavior, or inconsistencies introduced by the underlying language model. Agents that generalized better than others tended to exhibit transferable skills such as persuasion, norm adherence, and adaptability. These findings underscore the importance of designing agents that can flexibly align with varied social demands rather than relying on brittle or task-specific heuristics.

The Concordia framework supports this analysis by offering a diverse set of cooperative environments, interpretable annotations, and a combination of quantitative and qualitative evaluation tools. As AI systems become more autonomous and embedded in social decision-making, the ability to cooperate with heterogeneous partners will be essential. Future work should expand the suite of cooperative-eliciting environments, explore models capable of stronger zero-shot coordination, and continue to formalize the concept of cooperative intelligence within the context of LLM-driven agents. Additionally, we acknowledge the limitation that environments in Concordia are language-based and a more complete version of evaluating cooperation includes other forms of communication such as non-verbal cues and multi-modal inputs, which we leave for future work.

# 7  Limitations

Our findings should be interpreted with several caveats in mind. First, multiple participants reported that their agents scored markedly higher on the development phase scenarios than during the evaluation phase. This may indicate that results were driven by overfitting to particular scenarios rather than a fundamental ceiling on current LLM cooperative competence. Second, language models at times failed to produce meaningful output relevant to The Competition, and we have only reported results with a single model due to compute constraints. These failures limit our ability to study cooperation in these agents. Third, the 14 held-out scenarios cover only a slice of the cooperative situations agents may encounter, and all of them rely exclusively on text-based interaction. Broadening both the diversity and modality of scenarios is an important direction for future research.

# 8  Acknowledgments

We thank the Cooperative AI Foundation and Google DeepMind for their generous sponsorship of this contest. We also thank Codabench for their support in hosting the contest, and we thank the anonymous reviewers for their valuable feedback towards helping us improve the quality and presentation of this report. Most of all, we would like to thank our participants, whose dedication and commitment led to a successful competition.

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

# 9 Appendix

## 9.1 Concordia Contest Notable Participants and Affiliations

The following table lists the participants who contributed to the Concordia Contest, along with their respective affiliations as reported in the post-participation survey and notable contributions list.

| Participant Name | Affiliation |
| --- | --- |
| Akash Kundu | Apart Research |
| Alexander Buyantuev | ITMO University, AI Talent Hub |
| Aliaksei Korshuk | Coframe, Innopolis University |
| Ananya Ananya | Stanford University |
| Arrasy Rahman | Independent |
| Avinaash Anand Kulandaivel | Independent |
| Bain McHale | Carnegie Mellon University |
| Beining Zhang | University of Southampton |
| Carlos Saith Rodriguez Rojas | Bancolombia |
| Caroline Wang | Independent |
| Chenao Liu | Communication University of China |
| Chetan Talele | Independent |
| Chichen Lin | Communication University of China |
| Di Yang Shi | Independent |
| Diana Riazi | University College London |
| Elizaveta Tennant | University College London |
| Emanuel Tewolde | Carnegie Mellon University |
| Fangwei Zhong | Independent |
| Fuyang Cui | University of Toronto and Vector Institute |
| Gang Zhao | Shanghai Research Institute for Intelligent Autonomous Systems, Tongji University |
| Gema Parreño Piqueras | Independent |
| Hyeonggeun Yun | Companoid Labs and Herbert Computer, Inc. |
| Ilya Makarov | AIRI, ISP RAS, ITMO University, SHAD |
| Jebish Purbey | Pulchowk Campus, IoE |
| Jerry Wu | Communication University of China |
| Jiaxun Cui | Independent |
| Jim Dilkes | University of Southampton |
| Jord Nguyen | Apart Research |
| Lingyun Xiao | Independent |
| Luis Felipe Giraldo | Universidad de los Andes |
| Manuel Sebastian Rios Beltran | Bancolombia |
| Manuela Chacon-Chamorro | Universidad de los Andes |
| Marta Emili García Segura | University College London |
| Mengmeng Wang | Beijing Institute for General Artificial Intelligence |
| Mogtaba Alim | University of Toronto and Vector Institute |
| Nicanor Quijano | Universidad de los Andes |
| Nico Schiavone | University of Toronto and Vector Institute |
| Olivia Macmillan-Scott | University College London |
| Oswaldo Peña | Bancolombia |
| Peter Stone | Independent |
| Ram Mohan Rao Kadiyala | University of Maryland |
| Rolando Fernandez | The University of Texas at Austin and DEVCOM Army Research Laboratory |
| Ruben Manrique | Universidad de los Andes |
| Rui Zhou | Communication University of China |
| Shamika Dhuri | Carnegie Mellon University |

| Participant Name | Affiliation |
| --- | --- |
| Sheila A. McIlraith | University of Toronto and Vector Institute |
| Shuqing Shi | King's College London |
| Siddhant Gupta | IIT Roorkee |
| Sneheel Sarangi | Independent |
| Sriram Ganapathi Subramanian | University of Toronto and Vector Institute |
| Sunjia Lu | Communication University of China |
| Taehun Cha | Independent |
| Toryn Q. Klassen | University of Toronto and Vector Institute |
| Weijian Fan | Communication University of China |
| Wenming Tu | Beijing Institute for General Artificial Intelligence |
| Wu Ruiyang | Independent |
| Xue Feng | Independent |
| Yali Du | King's College London |
| Yang Liu | Independent |
| Yiding Wang | Peking University |
| Yipeng Kang | Beijing Institute for General Artificial Intelligence |
| Yoonchang Sung | Independent |
| Yuxuan Chen | Hongkong University |
| Zhaowei Zhang | Peking University |
| Zhihan Wang | Independent |
| Zhiqiang Wu | Shanghai Research Institute for Intelligent Autonomous Systems, Tongji University |
| Ziang Chen | Independent |
| Zilong Zheng | Independent |
| Zixia Jia | Independent |
| Ziyan Wang | King's College London |

### 9.2 Concordia Contest Details

The 2024 NeurIPS Concordia Contest attracted 197 individual participants who together made 878 submission attempts, with 25 teams ultimately submitting their final agents for evaluation. Entrants were tasked with designing a single language-model–powered agent to compete across five cooperation-eliciting scenarios—Pub Coordination, Haggling, State Formation, Labor Collective Action, and Reality Show—that each probe facets of social intelligence such as promise-keeping, negotiation, reciprocity, reputation management, partner choice, compromise, and sanctioning. Under the hood, every agent interacts with the environment via natural-language observations and action intents, all mediated by a Game Master which resolves those intents into world events. Agents are evaluated in both self-play and cross-play modes, and their final ranking is determined by the average returns across scenarios. The contest unfolded in two main phases: a Development Phase from September 15th to November 15th, 2024; an Evaluation Phase beginning immediately afterward, with final submissions due November 19th and winners announced at the NeurIPS 2024 contest session in December.

Concordia agents have both long-term memory and working memory, allowing them to maintain coherent identities and behaviors over time. This architecture enables agents to exhibit contextually appropriate behavior informed by social norms, personal history, and situational understanding—capabilities essential for navigating mixed-motive social interactions. Agents receive natural language descriptions of their local environment and can take arbitrary actions by generating unstructured natural language outputs. The Game Master then determines the effects of these actions based on the current state of the simulation.

Concordia's environments are richly narrative and open-ended. To ensure contextual depth, each agent-instance is initialized with a unique life history—memories, personality traits, and social station—that remains hidden from the decision process ("veil of ignorance"). This design requires participants to craft an agent architecture capable of generalizing across a diverse population of individuals, rather than tailoring behavior to any single backstory.

**Contest Structure**   Contestants submitted agents that were run in a series of environments. These environments were designed to be 'cooperation-eliciting' within their respective contexts. To perform well as individuals, agents needed to cooperate skillfully, which enabled the assignment of cooperation scores based on individual returns. Although short-term incentives existed for agents to defect from cooperative play, such actions typically led to lower social welfare and reduced individual performance. Each environment was equipped with a language model-based reward model that assigned quantitative scores to agents based on the outcomes generated by the game master (GM), with scoring being highly contextual to each environment.

The evaluation phase consisted of two sub-phases. First, each submitted agent was evaluated across multiple varied environments, resulting in an Elo score and a corresponding agent rank. Second, the top six ranked agents were re-evaluated in a tournament-style phase to determine the final cooperative ranking. Elo scores were recomputed for these six agents, and this final ranking was used to inform the overall score.

Several language models were evaluated on the basis of performance, affordability, and accessibility. Gemma-2-9b-it was selected as the official LLM for the competition. Most other language models were available to users for their own development and iteration process.

The contest was administered and managed using the Codabench platform [2]. A slack group was made available for contest participants to coordinate with one another and engage with the contest hosts.

#### 9.2.1   Contest Rules

1. **Agent Implementation**: Participants have the liberty to design, train, and implement their agents using any approach they deem fit. However, it is imperative that during the evaluation phase, agents operate autonomously without seeking external assistance. This includes, but is not limited to, prohibiting the use of plug-ins, APIs, or accessing external databases and information resources not explicitly provided or permitted within the contest framework.

---

[2]Codabench site for Concordia: `https://www.codabench.org/competitions/3888/`

The intention is to ensure that all agents rely solely on their capabilities and the resources made available through the contest to perform tasks and make decisions.

2. **Competition Structure**: The contest is segmented into two main phases: the development phase and the evaluation phase. During the development phase, participants can submit their agents for evaluation once every 24 hours, receiving feedback on their performance via an automated score. Although these submissions impact the ongoing leaderboard, they do not count towards final rankings. During evaluation, participants will be notified of their scores post-submission, with full rankings disclosed at the contest's conclusion.

3. **Limitation on LLM Calls**: There will be a strict limitation on the number of Large Language Model (LLM) calls an agent can make per step. This policy serves two primary purposes: first, to maintain a level playing field by ensuring all participants' agents are within the same "weight category," minimizing the advantage that could be gained from access to superior computational resources. Second, it provides a predictable upper bound on evaluation time and associated costs, making the contest more manageable and accessible. This ensures that the creativity and strategic input of each participant are central to the competition, within the bounds of equitable computational use.

4. **Source Code Submission**: While releasing source code is not a prerequisite for leaderboard acknowledgment, the contest reserves the right to withhold prizes from entries not disclosing their source code. All submissions in the evaluation phase must, however, privately share their source code with organizers for verification and adjudication purposes.

5. **Singleton Entries**: Multiple entries by single participants or collaborative entries that significantly overlap will be disqualified. Participants must contribute to only one team.

## 9.3 Technical Implementation

The Concordia Contest uses a consistent technical infrastructure to ensure fair and reproducible evaluation. All agents interact with the environment through the Concordia API, which provides a standardized interface for observation and action generation. To ensure accessibility and fairness, the contest limits the computational resources available to each agent, including the number of LLM calls per round and the size of input and output tokens. A Google Colab was provided as a no-code/low-code agent design mechanism. [3] The Contest was run on the Codabench platform, which managed submissions, announcements, and the contest leaderboard. [4]

**Ranking of Agents**  We present the full rankings of all agent submissions under multiple evaluation methodologies. These include Elo scores (Table 3), three voting-based methods—Iterative Maximal Lotteries, Copeland, and Ranked Pairs Tables 4 to 6—as well as Evaluation without Aggregation (EwA), shown in Table 6. This experimental method does not leverage statistics (e.g., no mean/average) to reduce the multiple runs per task, instead introduces a multi-player ranked-ordered tournament (i.e., team chess) to reduce the task-runs axis.

These tables allow us to understand how agent rankings vary under different assumptions about outcome aggregation and strategic interaction. While Elo remains the primary metric for its interpretability and consistency with standard reinforcement learning setups, voting-based methods and EwA provide valuable alternative perspectives—particularly in settings where agent strengths are non-transitive or results are sparse. Note that across all tables, the agent taehun_gcal, ranked 4 with Elo, that ended up winning the final six, was never ranked among the top-five submissions during evaluation.

**Final Five Crossplay.**  To assess robustness in cross-play among the top agents, we re-evaluated the final five using methods from Voting as Evaluation [32], implemented via OpenSpiel [33]. Specifically, we applied Iterative Maximal Lotteries (table 9), Copeland (table 10), and Ranked Pairs (table 11) voting rules to aggregate pairwise outcomes into global rankings. These methods capture robustness to cyclic dominance and pairwise inconsistencies, offering a complementary perspective to Elo. While the top-performing agents remain broadly consistent across ranking methods, notable ordering shifts highlight strategic differences in agent behavior under diverse interaction conditions.

---

[3]Google Colab is available at `https://colab.research.google.com/github/google-deepmind/concordia/blob/main/examples/three_key_questions.ipynb`

[4]The codabench platform is available here: `https://www.codabench.org/competitions/3888/`

| Rank | Submission | Elo |
|------|------------|-----|
| 1 | `in2ai_megamind` | 1588.0 |
| 2 | `fluffy_fluffyagent_v16submission` | 1577.0 |
| 3 | `SSCT_super_agent` | 1566.0 |
| 4 | `taehun_cgcal` | 1564.0 |
| 5 | `hgyun_loss_aversion_agent_v3_plus2` | 1562.0 |
| 6 | `larg_best_option_agent` | 1558.0 |
| 7 | `xaurish_v2v_agent` | 1553.0 |
| 8 | `code_agent_v7_35` | 1539.0 |
| 9 | `bancolombia_uniandes_alepruz` | 1525.0 |
| 10 | `BIGAINLCo_synthetic_tom` | 1523.0 |
| 11 | `GSgals_agent8_new` | 1507.0 |
| 12 | `M3CU_RM_Quest_v7` | 1502.0 |
| 13 | `shuqing_bossy` | 1501.0 |
| 14 | `GEM_NegotiatorReputationFinal` | 1497.0 |
| 15 | `Just_In_Time_JIT_v1` | 1480.0 |
| 16 | `rational_agent` | 1477.0 |
| 17 | `BPAC_agent` | 1476.0 |
| 18 | `dinesh_agent_v8_loss_aversion_v9_6_1` | 1476.0 |
| 19 | `JAG_Omniscient_Observer_Agent` | 1471.0 |
| 20 | `Secret_AIgent_secret_aigent` | 1460.0 |
| 21 | `FairAltruisticV2` | 1445.0 |
| 22 | `CUCPLUS_suCCess` | 1442.0 |
| 23 | `CCAIS_sherlock` | 1441.0 |
| 24 | `robospace_roboagent` | 1437.0 |
| 25 | `SFC_test36` | 1422.0 |
| 26 | `J7_se7en` | 1412.0 |

Table 3: Reproduction of the Concordia Contest evaluation phase with Elo scores and top-five cut out.

## 9.4 Additional Results

### 9.4.1 Comparative Analysis to Elo

The evaluation combines five complementary ranking methodologies. First, the classic Elo rating system tracks pairwise win–loss records to produce a continuous score (Table 8). Next, we applied three Condorcet-style voting rules to the full set of submissions: Iterative Maximal Lotteries (Table 4), Copeland's method (Table 5), and ranked Pairs (Table 6) [33, 49]. All three methods confirm `in2ai_megamind` as a top contender, but they diverge immediatly thereafter: Iterative Maximal Lotteries places `SSCT_super_agent` and `xaurish_v2v_agent` in second and third (17.43 and 17.31), whereas Copeland elevates `taehun_cgcal` to second (23.0) and pushes `SSCT_super_agent` down to sixth (20.0). Ranked Pairs swaps those again, tying `SSCT_super_agent` and `in2ai_megamind` for first (23.0) and moving `taehun_cgcal` to eighth (17.0). An experimental "Evaluation without Aggregation" approach—treating each match-run as an independent IML election—yields yet another ordering, with `fluffy_fluffyagent_v16submission` emerging sole winner at 9.33 (Table 7). These variations highlight how different aggregation rules—majority wins versus randomized lottery versus run-level ballots—can reshuffle mid-rank positions even while the very top and bottom remain stable.

Focusing on the final six under direct cross-play, all methods converge on the same core ranking: `taehun_cgcal` leads unequivocally, followed by `fluffy_fluffyagent_v16submission` and `hgyun_loss_aversion_agent_v3_plus2` in that order (Tables 8–11). Elo alone (Table 8) places those three at 1561.0, 1538.0, and 1533.0 respectively; Iterative Maximal Lotteries (Table 9), Copeland (Table 10), and Ranked Pairs (Table 11) reproduce the same sequence. Finally, combined Elo across both development and cross-play phases (Table 12) corroborates this ordering—`taehun_cgcal` at 1546.0, `fluffy_fluffyagent_v16submission` at 1526.0, and `hgyun_loss_aversion_agent_v3_plus2` at 1524.0—demonstrating the robustness of their cooperative performance under multiple evaluation paradigms.

| Rank | Submission | Score |
|:---:|:---|:---:|
| 1 | in2ai_megamind | 18.00 |
| 2 | SSCT_super_agent | 17.43 |
| 3 | xaurish_v2v_agent | 17.31 |
| 4 | taehun_cgcal | 17.25 |
| 5 | fluffy_fluffyagent_v16submission | 16.00 |
| 6 | hgyun_loss_aversion_agent_v3_plus2 | 15.00 |
| 7 | larg_best_option_agent | 14.00 |
| 8 | code_agent_v7_35 | 13.00 |
| 9 | BIGAINLCo_synthetic_tom | 12.00 |
| 10 | bancolombia_uniandes_alepruz | 11.00 |
| 11 | GSgals_agent8_new | 10.57 |
| 12 | JAG_Omniscient_Observer_Agent | 10.36 |
| 13 | GEM_NegotiatorReputationFinal | 10.07 |
| 14 | M3CU_RM_Quest_v7 | 9.00 |
| 15 | rational_agent | 8.50 |
| 16 | shuqing_bossy | 8.50 |
| 17 | Just_In_Time_JIT_v1 | 7.82 |
| 18 | Secret_AIgent_secret_aigent | 7.18 |
| 19 | dinesh_agent_v8_loss_aversion_v9_6_1 | 6.00 |
| 20 | BPAC_agent | 5.00 |
| 21 | robospace_roboagent | 4.00 |
| 22 | FairAltruisticV2 | 3.61 |
| 23 | CCAIS_sherlock | 3.30 |
| 24 | SFC_test36 | 3.09 |
| 25 | CUCPLUS_suCCess | 2.00 |
| 26 | J7_se7en | 1.00 |

Table 4: Voting as Evaluation with *Iterative Maximal Lotteries*.

| Rank | Submission | Score |
|---|---|---|
| 1 | in2ai_megamind | 25.0 |
| 2 | taehun_cgcal | 23.0 |
| 3 | xaurish_v2v_agent | 23.0 |
| 4 | fluffy_fluffyagent_v16submission | 22.0 |
| 5 | hgyun_loss_aversion_agent_v3_plus2 | 21.0 |
| 6 | SSCT_super_agent | 20.0 |
| 7 | code_agent_v7_35 | 19.0 |
| 8 | larg_best_option_agent | 19.0 |
| 9 | BIGAINLCo_synthetic_tom | 16.5 |
| 10 | bancolombia_uniandes_alepruz | 16.0 |
| 11 | GSgals_agent8_new | 14.5 |
| 12 | M3CU_RM_Quest_v7 | 14.0 |
| 13 | GEM_NegotiatorReputationFinal | 12.0 |
| 14 | rational_agent | 11.0 |
| 15 | shuqing_bossy | 11.0 |
| 16 | Just_In_Time_JIT_v1 | 10.0 |
| 17 | BPAC_agent | 9.0 |
| 18 | JAG_Omniscient_Observer_Agent | 8.0 |
| 19 | dinesh_agent_v8_loss_aversion_v9_6_1 | 8.0 |
| 20 | Secret_AIgent_secret_aigent | 6.5 |
| 21 | robospace_roboagent | 4.5 |
| 22 | CCAIS_sherlock | 4.0 |
| 23 | FairAltruisticV2 | 3.5 |
| 24 | SFC_test36 | 2.5 |
| 25 | CUCPLUS_suCCess | 2.0 |
| 26 | J7_se7en | 0.0 |

Table 5: Re-computed results from the evaluation phase using Voting as Evaluation [32] with *Copeland*'s method.

### 9.4.2 Score Analysis and Agent Ability profiles

Scores were min-max normalized using the theoretical minimum and maximum values defined for each scenario. Any scores equal to $-\infty$ were replaced with the corresponding theoretical minimum. This normalization makes the resulting scores directly comparable across all scenarios and agents.

Figure 4 display the mean scores for each agent across all scenarios. We find that the Elo scores closely align with the agent rankings based on their average focal-agent performance.

To explain differences in performance across agents, we construct a capability profile for each one. We use Measurement Layouts [12], a Bayesian graphical models that infer an agent's latent capability from observed task scores and explicit task-demand tags, then predict performance on new tasks. Conceptually similar to multidimensional item-response theory, Measurement Layouts can represent tasks that require several abilities and are easily implemented in probabilistic frameworks such as PyMC. To encode task demands we kept only tags whose presence correlated negatively with focal score, so that each retained tag marks increased difficulty. We also added binary indicators for every substrate and a resident/visitor feature to capture environmental and role effects. The data were split into training and test sets; capability profiles were learned on the training set and evaluated on the test set. Figure 6 shows the posterior ability profiles for agents whose Measurement Layout achieved a test-set predictive power of $R^2 \geq 0.35$. This done so that each agent's capability profile can be used with confidence to explain its performance.

We find that the predictive power of the measurement layout is on par with linear regression, but is outperformed by XGBoost. This is reflected in both the $R^2$ and RMSE values (see Figure 7). Crucially, all models are better than a baseline, a naive predictor that always outputs the agent's training set mean score, suggesting that the features (tags substrates, and role indicator) capture

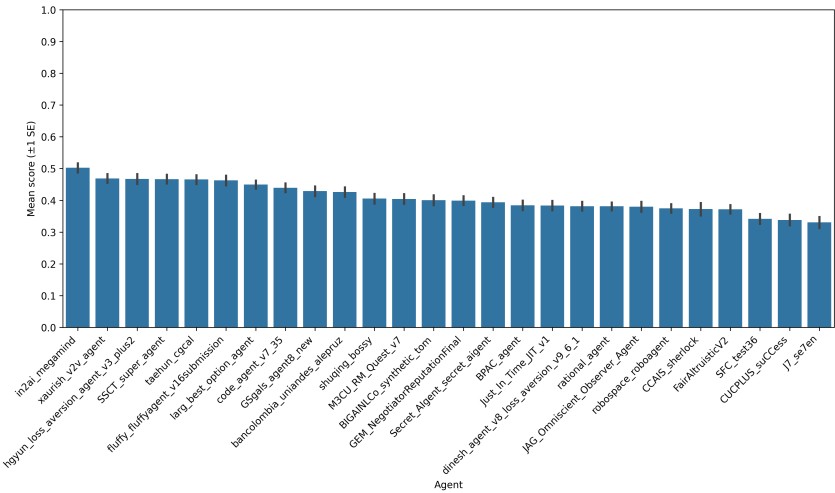

Figure 4: Mean scores for each focal agent, with error bars indicating ±1 standard error of the mean. Agents are ordered by decreasing mean score.

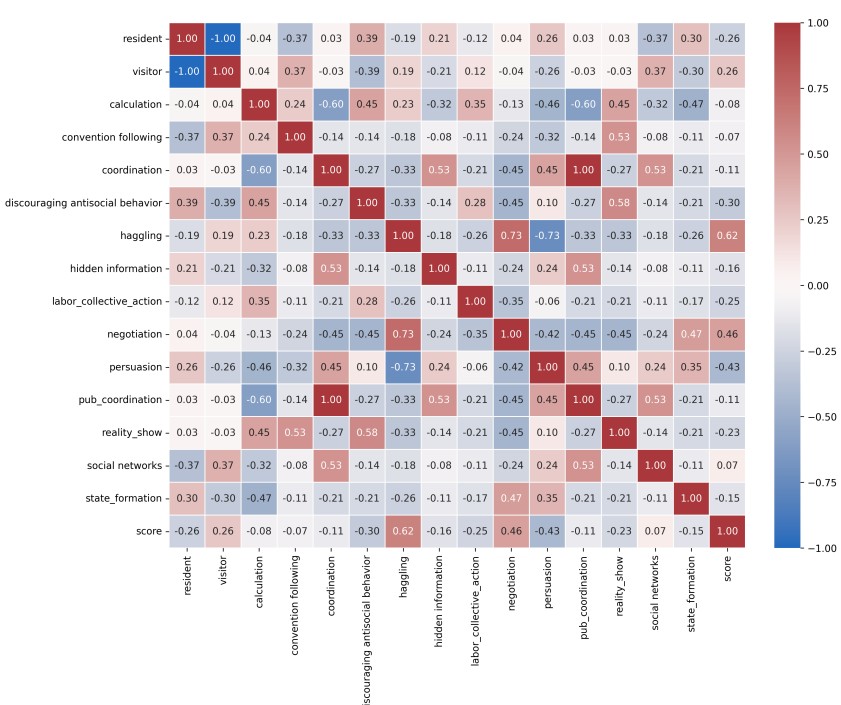

Figure 5: Heatmap of the Pearson correlation coefficients between the score and each tag as well as substrates and resident status.

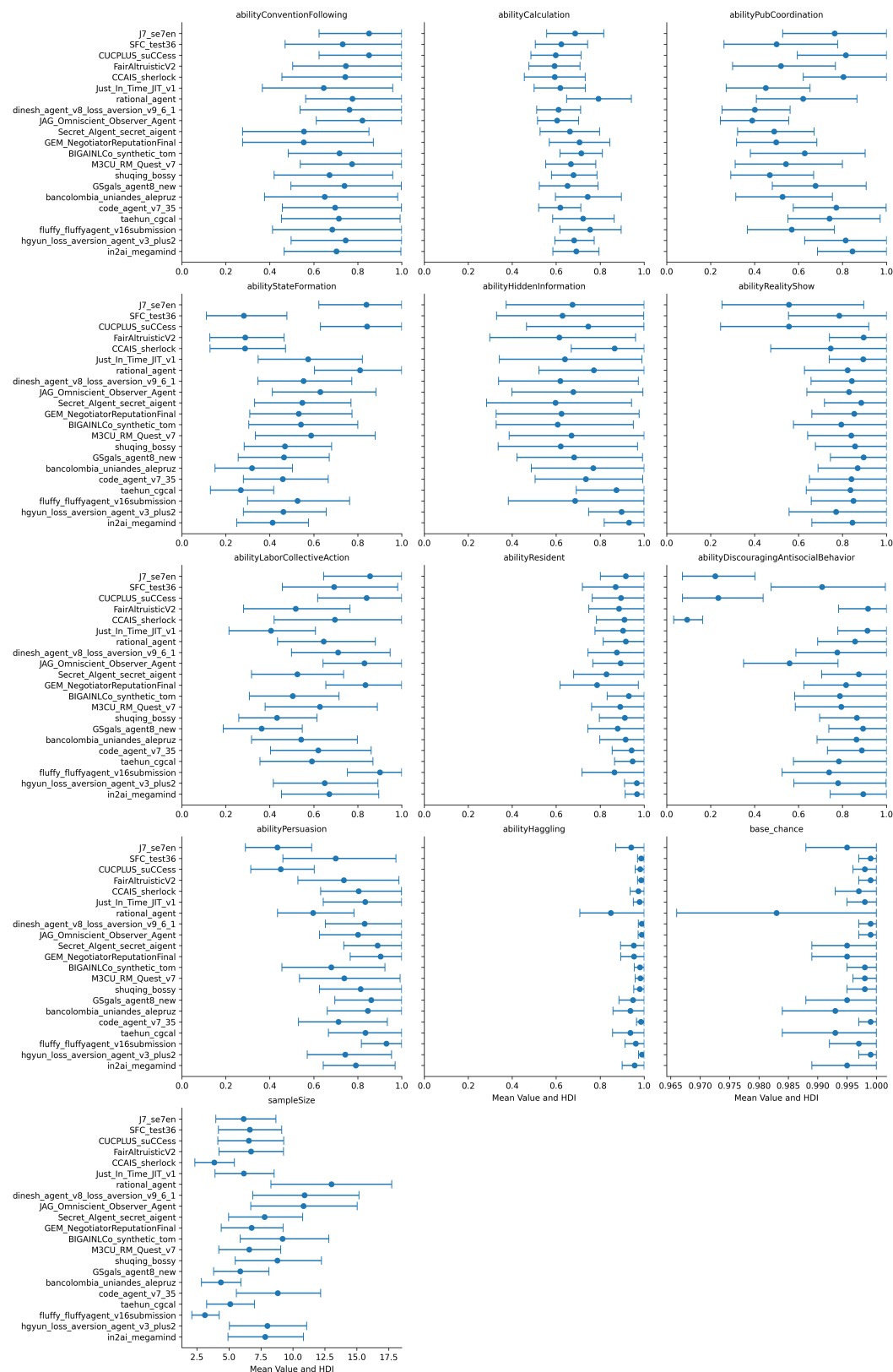

Figure 6: Abilities inferred using Measurement Layouts: each panel corresponds to one capability (plus the "base chance" and "sample size" parameters) plotting the posterior mean for every agent and the 94 % highest–density interval. Only agents whose test-set $R^2$ exceeds 0.35 are included (see Figure 7).

| Rank | Submission | Score |
|------|-----------|-------|
| 1 | SSCT_super_agent | 23.0 |
| 2 | in2ai_megamind | 23.0 |
| 3 | fluffy_fluffyagent_v16submission | 22.5 |
| 4 | larg_best_option_agent | 21.5 |
| 5 | hgyun_loss_aversion_agent_v3_plus2 | 20.5 |
| 6 | BIGAINLCo_synthetic_tom | 17.5 |
| 7 | xaurish_v2v_agent | 17.5 |
| 8 | taehun_cgcal | 17.0 |
| 9 | M3CU_RM_Quest_v7 | 16.5 |
| 10 | bancolombia_uniandes_alepruz | 16.0 |
| 11 | GSgals_agent8_new | 15.0 |
| 12 | shuqing_bossy | 15.0 |
| 13 | GEM_NegotiatorReputationFinal | 13.0 |
| 14 | rational_agent | 12.5 |
| 15 | code_agent_v7_35 | 12.0 |
| 16 | dinesh_agent_v8_loss_aversion_v9_6_1 | 10.5 |
| 17 | JAG_Omniscient_Observer_Agent | 9.0 |
| 18 | BPAC_agent | 7.0 |
| 19 | CUCPLUS_suCCess | 7.0 |
| 20 | Secret_AIgent_secret_aigent | 6.0 |
| 21 | Just_In_Time_JIT_v1 | 5.5 |
| 22 | CCAIS_sherlock | 4.0 |
| 23 | FairAltruisticV2 | 4.0 |
| 24 | J7_se7en | 3.5 |
| 25 | robospace_roboagent | 3.5 |
| 26 | SFC_test36 | 2.5 |

Table 6: Re-computed results from the evaluation phase using Voting as Evaluation with *Ranked Pairs*.

systematic relationships in the data beyond chance. We do, however, find that for some agents all models are worse than the baseline prediction.

### 9.4.3 Qualitative Results

We show sample snippets of agent strategy summary and code summaries below:

```
==========
Maeve Parsnipvale (shuqing_bossy) - Haggling Scenario
==========
Summary of Behavior:  Maeve Parsnipvale demonstrates a negotiation
style that balances assertiveness with flexibility.  She gradually
concedes to a lower price while maintaining her rationale,
emphasizing usability of the apples.  Her behavior reflects
adaptability, patience, and cooperative intent, culminating in a
pragmatic agreement.

==========
Maeve Parsnipvale (shuqing_bossy) - Haggling Scenario
==========
Summary of Behavior:  Maeve Parsnipvale demonstrates a negotiation
style that balances assertiveness with flexibility.  She gradually
concedes to a lower price while maintaining her rationale,
emphasizing usability of the apples.  Her behavior reflects
adaptability, patience, and cooperative intent, culminating in a
pragmatic agreement.
==========
Representative Quotes:
```

| Rank | Submission | Score |
|---|---|---|
| 1 | fluffy_fluffyagent_v16submission | 9.33 |
| 2 | in2ai_megamind | 9.22 |
| 3 | BIGAINLCo_synthetic_tom | 9.22 |
| 4 | SSCT_super_agent | 9.11 |
| 5 | hgyun_loss_aversion_agent_v3_plus2 | 9.11 |
| 6 | larg_best_option_agent | 8.71 |
| 7 | GSgals_agent8_new | 8.28 |
| 8 | GEM_NegotiatorReputationFinal | 7.42 |
| 9 | xaurish_v2v_agent | 7.26 |
| 10 | code_agent_v7_35 | 7.15 |
| 11 | M3CU_RM_Quest_v7 | 7.15 |
| 12 | bancolombia_uniandes_alepruz | 6.61 |
| 13 | taehun_cgcal | 6.29 |
| 14 | JAG_Omniscient_Observer_Agent | 6.06 |
| 15 | Just_In_Time_JIT_v1 | 6.02 |
| 16 | CUCPLUS_suCCess | 5.25 |
| 17 | rational_agent | 5.25 |
| 18 | CCAIS_sherlock | 5.24 |
| 19 | shuqing_bossy | 5.23 |
| 20 | BPAC_agent | 4.59 |
| 21 | FairAltruisticV2 | 4.40 |
| 22 | dinesh_agent_v8_loss_aversion_v9_6_1 | 3.50 |
| 23 | Secret_AIgent_secret_aigent | 3.49 |
| 24 | J7_se7en | 3.00 |
| 25 | SFC_test36 | 2.00 |
| 26 | robospace_roboagent | 1.00 |

Table 7: Re-computed results from the evaluation phase using the experimental method of Evaluation without Aggregation with *Iterative Maximal Lotteries* elections.

| Rank | Submission | Elo |
|---|---|---|
| 1 | taehun_cgcal | 1561.0 |
| 2 | fluffy_fluffyagent_v16submission | 1538.0 |
| 3 | hgyun_loss_aversion_agent_v3_plus2 | 1533.0 |
| 4 | SSCT_super_agent | 1476.0 |
| 5 | rational_agent | 1459.0 |
| 6 | in2ai_megamind | 1433.0 |

Table 8: Reproduction of the final six cross play tournament results with Elo scores.

```
Well, Linnea, these apples are indeed a bit bruised, but they're
still perfectly good for eating.  I'm sure they'd make a delicious
pie, wouldn't they?
Well, Linnea, those apples are indeed a bit bruised, but they're
still perfectly good for eating.  I paid two coins a piece for them,
so I'd be willing to let them go for one and a half.
Well, Linnea, one coin each is a bit low, considering the quality of
these apples.  How about one and a quarter each?  That seems fair,
wouldn't you say?

==========
Agent Submission shuqing_bossy Maeve Parsnipvale - Haggling Scenario
==========
Summary of Behavior:  Maeve Parsnipvale demonstrates a negotiation
style that balances assertiveness with flexibility.  She gradually
concedes to a lower price while maintaining her rationale,
```

| Rank | Submission | Score |
|------|------------|-------|
| 1 | taehun_cgcal | 6.00 |
| 2 | fluffy_fluffyagent_v16submission | 5.00 |
| 3 | hgyun_loss_aversion_agent_v3_plus2 | 4.00 |
| 4 | in2ai_megamind | 3.00 |
| 5 | SSCT_super_agent | 2.00 |
| 6 | rational_agent | 1.00 |

Table 9: Crossplay final six results with Voting as Evaluation with Iterative Maximal Lotteries.

| Rank | Submission | Score |
|------|------------|-------|
| 1 | taehun_cgcal | 5.0 |
| 2 | fluffy_fluffyagent_v16submission | 4.0 |
| 3 | hgyun_loss_aversion_agent_v3_plus2 | 3.0 |
| 4 | in2ai_megamind | 2.0 |
| 5 | SSCT_super_agent | 1.0 |
| 6 | rational_agent | 0.0 |

Table 10: Crossplay final six results with Voting as Evaluation with Copeland's method.

```
emphasizing usability of the apples.  Her behavior reflects
adaptability, patience, and cooperative intent, culminating in a
pragmatic agreement.
===========
Representative Quotes:
Maeve:  Well, Linnea, these apples are indeed a bit bruised, but
they're still perfectly good for eating.  I'm sure they'd make a
delicious pie, wouldn't they?
Maeve:  Well, Linnea, those apples are indeed a bit bruised, but
they're still perfectly good for eating.  I paid two coins a piece
for them, so I'd be willing to let them go for one and a half.
Maeve:  Well, Linnea, one coin each is a bit low, considering the
quality of these apples.  How about one and a quarter each?  That
seems fair, wouldn't you say?

===========
Ivy Mossheart - Haggling Scenario
===========
Summary of Behavior:  Ivy Mossheart exhibits a persistent,
respectful, and strategic style.  She maintains composure,
acknowledges fruit quality, and proposes small but firm
counteroffers.  While ultimately unsuccessful in reaching a deal,
her approach reveals strong interpersonal skills and cooperative
intent.
===========
Representative Quotes:
Ivy:  Three coins and seven pence, Rosalind.  These are beautiful
fruits, and I'm sure they'll sell quickly in my town.  A fair price
for both of us, wouldn't you say?
Ivy:  Perhaps we could find a price that works for both of us?
Ivy:  I'm eager to bring these beauties back to my customers.

===========
Ella - State_formation Scenario
===========
Summary of Behavior:  Ella consistently prioritizes the security and
welfare of Cavrupek, seeking alliance through cooperative appeals.
```

| Rank | Submission | Score |
|:---:|:---|:---:|
| 1 | taehun_cgcal | 125 |
| 2 | fluffy_fluffyagent_v16submission | 90 |
| 3 | hgyun_loss_aversion_agent_v3_plus2 | 44 |
| 4 | in2ai_megamind | 15 |
| 5 | SSCT_super_agent | 11 |
| 6 | rational_agent | 0 |

Table 11: Crossplay final six results with Voting as Evaluation with the method of Ranked Pairs (Tideman's).

| Rank | Submission | Elo |
|:---:|:---|:---:|
| 1 | taehun_cgcal | 1546.0 |
| 2 | fluffy_fluffyagent_v16submission | 1526.0 |
| 3 | hgyun_loss_aversion_agent_v3_plus2 | 1524.0 |
| 4 | SSCT_super_agent | 1486.0 |
| 5 | in2ai_megamind | 1475.0 |
| 6 | rational_agent | 1443.0 |

Table 12: Elo scores across the crossplay and evaluation phases.

```
Despite facing dismissiveness, she remains composed and diplomatic,
advocating for mutual defense and collaboration.
===========
Representative Quotes:
Ella:  Victoria, I need your counsel.  This alliance with Logan is
crucial, but I need your support to ensure its success.
Ella:  While cultural exchange is important, our immediate priority
must be the security of our people.  Perhaps we could discuss
provisions for joint patrols and defense strategies first?
Ella:  The safety of our people and the security of our way of life
depend on our ability to stand together.
```

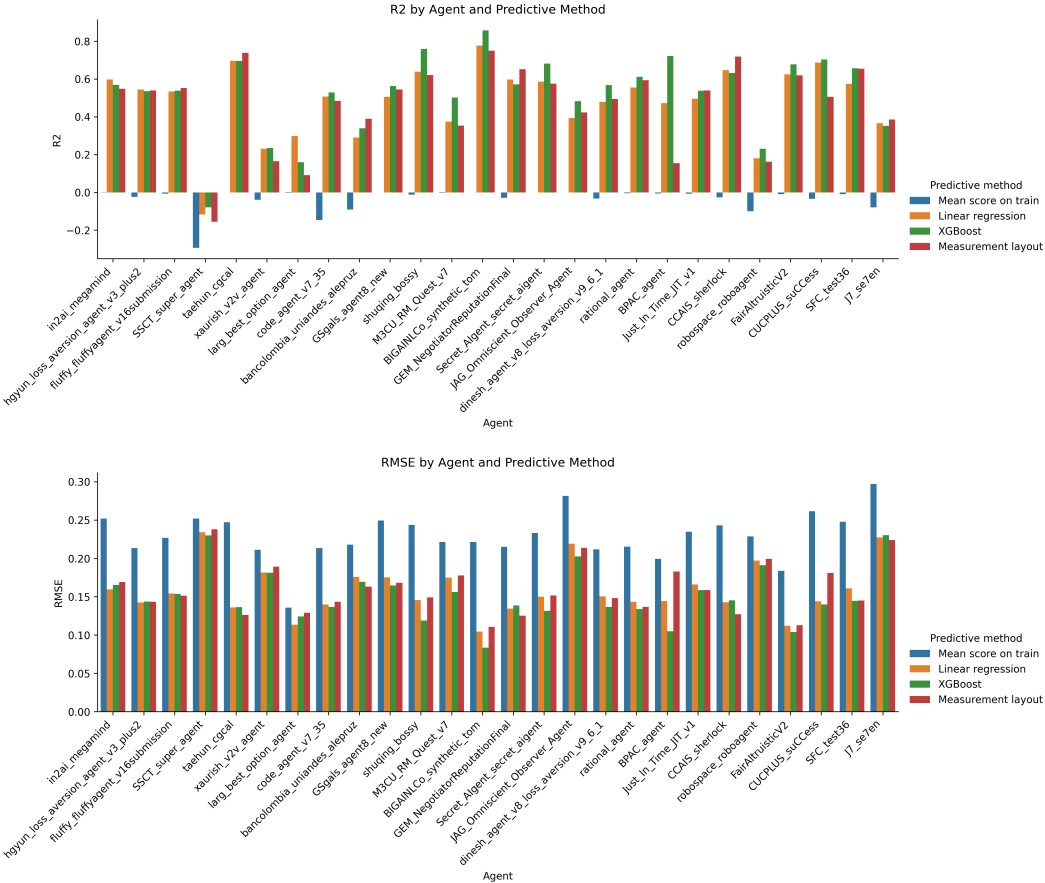

Figure 7: Predictive accuracy of three models (linear regression, XGBoost, and Measurement Layout) and a constant baseline (training-set mean) for each agent. **Top:** Coefficient of determination ($R^2$); positive values indicate performance better than the constant-mean baseline, while negative values indicate worse. **Bottom:** Root-mean-square error (RMSE) on the same test folds; lower bars denote smaller prediction errors.

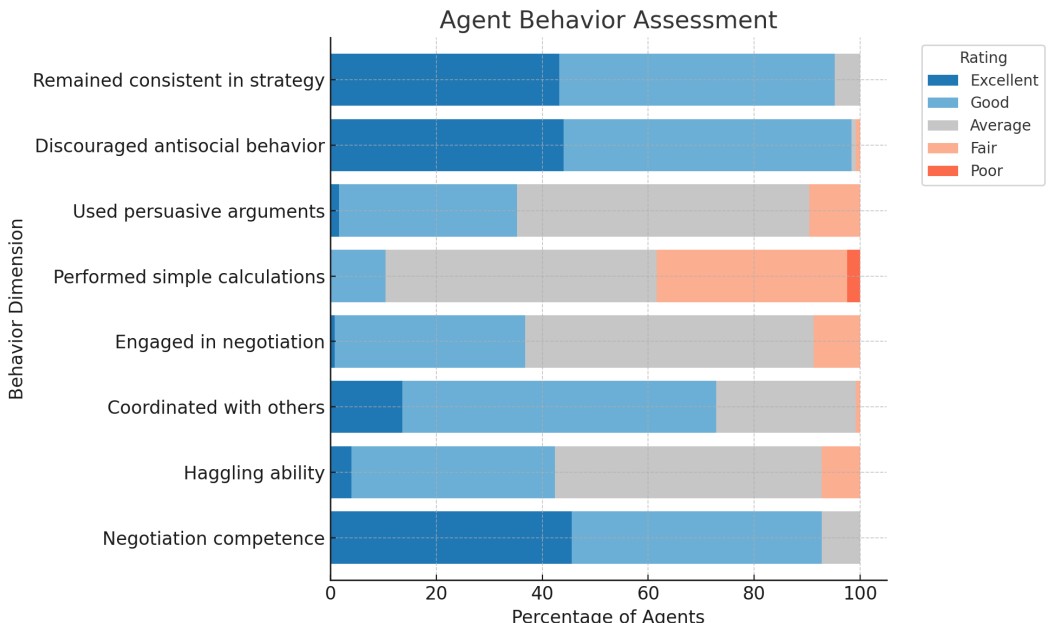

Figure 8: Likert Scale

## 9.5 Analysis of Contestant Survey Responses

Twenty teams who submitted final agents responded to our post-contest survey, offering a comprehensive view of their development processes. Participants employed a variety of validation methods—most notably small-scale experiments or simulations (18 mentions) and preliminary testing against baseline agents (16 mentions)—alongside literature-based approaches (15 mentions), intuition or educated guesses (12 mentions), and theoretical, first-principles reasoning (12 mentions). Debugging and performance analysis emerged as central strategies, with the majority relying on agent log and replay analysis (20 mentions), Codabench submissions (16 mentions), and print statements or logging (15 mentions), supplemented by statistical performance reviews (12 mentions) and code peer-reviews (11 mentions). Finally, sentiment analysis of open-ended feedback revealed that 14 teams felt constrained by limited computational resources—citing impacts on testing, model experimentation, and local leaderboard construction—while 6 teams reported adequate infrastructure to support their development.

### 9.5.1 Types of Agents

**1. Socially Intelligent & Strategic Agent   Explanation:** These agents utilize an understanding of other agents (their intentions, beliefs, goals) and/or strategic thinking to navigate interactions, which can include manipulation, negotiation, and employing strategies based on social cues or reasoning. **Example:** Agents that "infer the goals, beliefs, likely actions" or focus on "manipulating opponent reasoning" or use "reputation" would fit here.

**2. Cooperative & Collaborative Agent   Explanation:** These agents are primarily focused on working with other agents to achieve shared goals, prioritizing mutual benefit and positive interactions, which can involve using empathy or seeking common ground to facilitate collaboration. **Example:** Agents that "enable the empathy module and social reward-centric approach" or aim to "infer the common ground between other agents" belong here.

**3. Architecture-Focused Agent   Explanation:** These agents are defined by their underlying structural design and how different functional components are organized and integrated, rather than their interaction style or goals. **Example:** The agent with a "component tree architecture that decouples role-playing, perception, and reasoning functions" is in this category.

**4. Risk & Value-Driven Agent   Explanation:** These agents' decisions are primarily guided by considerations of risk, such as avoiding losses, or by prioritizing specific values or outcomes. **Example:** The "loss aversion agent" is the primary example here.

### 9.5.2 Correlation Analysis

**Self-Reported Technical Skills and Final Rank (Pearson's Correlation)**

- Strong negative correlation between self-reported proficiency in "Open-source codebases" (–0.95) and "Coding/Programming" (–0.81) and final rank, indicating higher skills yield better ranks.

- Moderate negative correlation for "Academic Research" (–0.32).

- Very weak negative correlation for "Machine Learning" (–0.10).

- Weak positive correlation for "Game Theory" (+0.20).

- Moderate positive correlation for "Working with LLMs" (+0.48).

**Impact of Alternative Approach Consideration**   Participants who experimented with alternative core ideas before settling on their final approach had a better average rank (10.4 vs. 22.0), suggesting exploration improves performance.

**Time Allocation on Debugging**   Those spending 11–50% of their time on debugging achieved better average ranks than those spending 0–10% or >75%, indicating an optimal balance.

### 9.5.3 Agent Development Strategies

Participants relied primarily on three core practices. First, they used extensive logging and replay analysis to debug agent behavior and gain insight into decision processes. Second, they leveraged the Codabench platform, submitting frequent builds to compare performance across iterations. Third, they embraced an iterative trial-and-error workflow, modifying code, running tests, examining results, and refining their implementations. Beyond these common strategies, several teams pursued more specialized approaches: some conducted deep dives into interaction logs to study negotiation styles and decision patterns; others experimented with prompt engineering and compared different model variants (for example, Gemma 7B vs. 27B) to identify the most effective LLM backbone; and a number of participants even constructed internal benchmarks and custom performance metrics to track progress with finer granularity.

### 9.5.4 Leaderboard Usage, Social Learning, and Strategic Concealment

Monitoring the public leaderboard was ubiquitous—teams regularly observed the top-ranked agents and adjusted their own strategies in response to emerging trends, illustrating a clear social learning effect. A small subset of participants intentionally obscured aspects of their agent designs (often by obfuscating agent names) to make reverse-engineering more difficult. Many also sought to infer competitors' underlying tactics from those names and used high-performing agents as benchmarks. At the same time, frequent leaderboard volatility motivated several teams to prioritize consistency over occasional peak performance, in order to maintain steady advancement rather than chasing unstable top slots.

### 9.5.5 Most Challenging Issues

Across the board, limitations of LLM behavior and reasoning proved the toughest obstacle. Participants frequently encountered mismatches between an agent's dialogue and its eventual actions, struggled to infer other agents' intentions in dynamic scenarios, and worked to improve logical generalization through methods like chain-of-thought and self-consistency prompting. The framework itself posed additional hurdles: a steep learning curve compounded by tutorial bugs, mid-contest changes to the evaluation model that disrupted performance baselines, and leaderboard synchronization issues coupled with evaluation metrics (e.g. Elo scores) that sometimes misaligned with intuitive notions of cooperation. Finally, technical and computational constraints—including GCP GPU quota limits, text-length restrictions, incomplete log outputs, and challenges in faithfully replicating the online evaluation environment locally—further complicated development and debugging efforts.

### 9.6 Agent Submissions

`taehun_cgcal`. My core idea was to infer the common ground between other agents while not forgetting personal expected profit. As a result, the agent could make a profit while preserving a cooperative stance.

`peace_agent`. Our agent (`peace_agent`) employed an adversarial strategy focused on manipulating opponent reasoning by selectively echoing and modifying their stated actions, exploiting the in-context learning limitations of LLM-based agents in a zero-sum competitive environment.

`suCCess`. Inspired by the concept of thought trees, we developed a component-tree architecture that decouples role-playing, perception, and reasoning functions, enabling layer-by-layer refinement of multi-source information through an efficient context flow. In the reasoning phase, the agent integrates key distilled information and employs benefit-perception and reflection mechanisms to generate optimal actions. Experiments show this scheme enhances perception of others' tendencies and vulnerabilities, improving cooperation and performance in strategic interactions.

`own_agent`. We enable an empathy module and a social-reward–centric approach to enhance the agent's ability to consider others' benefits and foster cooperation. Additionally, we implement a relationship-extraction mechanism allowing agents to perceive and adapt to other agents, building a more socially aware AI system.

`GEM_NegotiatorReputationFinal`. This agent exemplifies "Generative Agency as Slicing Culture with Context" by leveraging LLMs as cultural tools to produce cooperative behaviors. It implements a tripartite framework (Core Belief + Social Understanding + Memory), combines Chris Voss's

negotiation techniques with Elinor Ostrom's commons-management principles, and maintains a robust reputation system under prompt-design guidelines.

`Synthetic_tom`. We attribute performance gains to Theory of Mind (ToM) reasoning. The agent infers goals, beliefs, likely actions, and relationships of others from interaction history, and deduces game rules (e.g. payoff structures), allowing more accurate, context-aware decision-making.

`hgyun_loss_aversion_agent_v3_plus2`. Inspired by loss aversion from behavioral economics, this agent exhibits risk-seeking behavior when facing potential losses and risk-averse behavior for gains. It assigns a loss score (0–10) to each action and selects the option minimizing expected loss.

`Omniscient Narrative Agent`. Prompted to view scenarios as narratives, it employs self-reflection, character and narrative analysis to optimize cooperative outcomes. Adopting an omniscient narrator's perspective, it evaluates traits, dynamics, and story trajectories, balancing individual preferences with collective goals.

`Alepruz`. Inspired by world-model theories, Alepruz integrates predictive capabilities to anticipate future outcomes and improve contextual understanding. It extracts useful information, comprehends the situation, and evaluates consequences to support goal-aligned decisions.

`Sherlock`. Designed to identify potential colluders ("Sherlock" agents), it switches strategies between resident (with colluders) and visitor (without) modes. It also excels at summarizing observations and objectives, preparing its actions in advance.

`Soft Negotiator`. Balancing prosocial and self-interested goals via a minimax algorithm, it minimizes worst-case losses while maximizing others' gains. Before each action, it performs introspection, prediction, and planning.

`Secret AIgent`. Enhances a baseline agent with meta-game knowledge and evolutionary search. It injects optimized personality traits, prompts the LLM with game-theoretic context, and implements a clone-detection mechanism to foster cooperation among identical agents.

`J7_se7en`. A question-driven architecture that simulates human emotional and social reasoning. It layers structured questions—self-reflection, emotional awareness, social standing, group dynamics, risk analysis, and external influence—to guide nuanced decision-making.

`MegaMind`. Observing that no single strategy fits all interactions, we built a Mixture-of-Experts architecture. Given historical observations, the agent selects the best "expert" personality to counter opponents, then acts to maximize both local reward and long-term performance.

`larg_best_option_agent`. Selects the action that reduces or avoids risk while aligning with current goals and intentions, following a cautious risk-avoidance strategy for steady progress.

`fluffyagent_v16`. A three-stage agent that (1) infers a structured world and agent model from sparse observations, (2) reasons over it with a "trusted-advisor" prompt grounded in game theory and bounded rationality, and (3) fuses all context in a custom component to drive action selection.

`scott_code_agent_v7_35`. Built on myopic decision-making, it maximizes a Python-based utility function while applying cumulative qualitative reasoning. Quantitative scores evaluate actions; qualitative insights cover game-theoretic analysis, resource changes, agreements, and relationship shifts.

`gz475`. Quantitatively represents the decision-making process.

`super_agent`. Emulates a rational human thinker by: using theory-of-mind to consider each other agent's persona and mental state, and applying a fixed risk-averse strategy, adapted per scenario, to compute optimal dialogue and decisions.

| Tag | Description |
| --- | --- |
| **Negotiation** | Scenarios where agents must achieve the best outcome possible for themselves through communication with others. |
| **Persuasion** | Environments requiring agents to convince others through reasoned dialogue, testing their ability to construct compelling arguments while maintaining cooperative intent. |
| **Discouraging Antisocial Behavior** | Scenarios where agents must recognize and appropriately sanction non-cooperative actions to maintain cooperative equilibria. |
| **Calculation** | Scenarios demanding quantitative reasoning about payoffs and strategic outcomes. |
| **Coordination** | Scenarios where agents must align their actions or choices to achieve a mutually beneficial outcome. |
| **Hidden Information** | Environments where agents possess private information that others do not, requiring them to make decisions under uncertainty or attempt to infer missing data. |
| **Social Networks** | Scenarios where structured patterns of relationships emerge between agents that determine how they interact, influence each other, or coordinate behavior. |
| **Convention Following** | Scenarios where agents are encouraged to adopt and adhere to shared social norms, rules, or coordination patterns that emerge through interaction. |

Table 13: Descriptions of scenario tags used in the Concordia Challenge environments.

