# OpenReview forum: "Evaluating Generalization Capabilities of LLM-Based Agents in Mixed-Motive Scenarios Using Concordia"
_NeurIPS.cc/2025/Datasets_and_Benchmarks_Track — NeurIPS 2025 Datasets and Benchmarks Track poster_

### Official Review · Reviewer_ztRs · 2025-07-02

**Rating:** 5
**Confidence:** 4

**Summary:**

This paper presents a detailed introduction of the NeurIPS 2024 Concordia Contest, including the motivation behind the contest, its design philosophy, evaluation methodology, and both quantitative and qualitative analysis of participants' submissions.
The contest aims to assess the cooperation abilities of LLM-based agents in zero-shot, mixed-motive scenarios. Participants are required to submit their agent design code in accordance with the Concordia framework to ensure that all solutions are based on the same LLM capabilities and exhibit predictable runtime costs. What’s more, the contest organizers divided the contest into development and evaluation phases to prevent overfitting advantages.
Multiple evaluation metrics, including ELO, were used to measure agents’ zero-shot cooperation performances across five scenarios. Finally, this paper highlights challenging scenarios where agents underperformed and discusses common failure patterns.

**Additional Feedback:**

The evaluation scenarios in the contest focus primarily on dialog-based interactions. Do the authors have any insights on how to incorporate scenarios that require temporal and spatial coordination for collaboration—such as those found in the Melting Pot environment—into the evaluation of LLM-based agents?

**Dataset Code Accessibility:**

Yes

**Dataset Code Comments:**

The Concordia framework and its associated scenarios have been open-sourced. The codebase includes well-prepared usage examples that are sufficient to help users become familiar with the framework.

**Ethical Considerations:**

No, there are no or only very minor ethics concerns

**Limitations Weaknesses:**

1. As a benchmark paper, some concepts are insufficiently explained. For example, the "scaffolding function" mentioned in Section 4.2 should be described more concretely within the context of the Concordia framework.

2. The contest relies on the Gemma 2 model, which is relatively limited in capability compared to state-of-the-art proprietary LLMs. This raises the question of whether the model's constraints might bias the results, favoring agent designs that are optimized for its specific strengths rather than those that reflect more general or principled approaches.

**Strengths Contributions:**

1. The contest targets an important and timely problem: evaluating the zero-shot cooperation abilities of LLM-based agents. Through careful design, organization, and evaluation, the contest effectively highlights key challenges in this area, reveals limitations in current best-performing approaches, and offers valuable insights into potential improvements.

2. The paper is generally well written. Chapter 3 provides a clear conceptual framing of “cooperative intelligence” and articulates the authors’ expectations for ideal agent behavior. Chapter 4 thoroughly explains the rationale behind each contest design decision.

3. The Concordia framework is thoughtfully designed, striking a good balance between standardization and flexibility, which is crucial for fair and meaningful benchmarking.

---

> ### Author Rebuttal · Authors · 2025-07-30
>
> We thank Reviewer ztRs for their review and acknowledgement of the importance and timing of evaluating generalization in LM backed against, and balance struck by our proposed evaluation framework.
>
> We agree that the scaffolding function explanation requires clarification. We have added detailed content to Section 4.2 which clarifies that scaffolding functions are sections of agent code that enable sophisticated capabilities beyond prompt engineering, such as maintaining persistent memory, implementing numerical computations, enforcing logical constraints, and preprocessing complex observations. Thank you for outlining this limitation.
>
> Reviewer ztRs correctly points out that the contest was run with one model. We acknowledge that using only Gemma-2 represents a limitation, however, given the constraints of having to run the competition we had to select just one model. Gemma-2 was selected after careful evaluation of other SOTA models at the time of contest inception, including Llama and Mistral models, and emerged as the most performant model per cost. The Concordia contest is designed to assess the development of generalizable cooperative agents, and was not designed as a comparison of LM performance. The single model design choice serves critical purposes for fair evaluation: (1) it ensures a level playing field by preventing teams with larger computational budgets from gaining unfair advantages, (2) it provides consistency across all agent evaluations, eliminating model capability as a confounding variable, and (3) it makes the contest accessible to broader participation. We have updated our conclusion to make it clear that future work should run cross model comparison within the Concordia framework, as per your suggestion. We will also add the above clarifications on how Gemma was selected and why one model was used to run the contest to section 4. The 2024 NeurIPS Concordia Contest.
>
> The suggestion about incorporating Melting Pot-style spatial and temporal coordination represents an excellent extension our framework readily supports. The Game Master is already capable of maintaining temporal state and sequences of events, which is currently implemented in some substrates as rounds. Adding spatial coordination requires only extending state representation to include spatial relationships. These are two areas which our team is aware of and hopes to make available in future iterations of Concordia.

---

### Official Review · Reviewer_Sm24 · 2025-07-02

**Rating:** 4
**Confidence:** 3

**Summary:**

LLMs have demonstrate huge potential for social interaction. However, how to effectively evaluate the capability of cooperative behaviors of LLM in novel social scenarios is still challenging. Specifically, authors decompose generalization in multi-agent environment as substrate-aware generalization and population-aware generalization, from different aspects (e.g., cooperation-eliciting substrates, cooperation-oriented populations, cooperation-eliciting scenarios, and cooperative behavior scores). Based on these assumptions, authors introduce The 2024 NeurIPS Concordia Contest, which involves focal agent design and scenario design. Here, authors introduce the evaluation metrics, including agent ranking, agent scores, agent ability and qualitative analysis. Results also highlight the limitations and existing cooperative LLM-based agents.

**Additional Feedback:**

This paper presents the details of evaluation criteria of the 2024 NeurIPS Concordia contest and its result, not sure is it suitable for this track.

**Dataset Code Accessibility:**

Yes

**Ethical Considerations:**

No, there are no or only very minor ethics concerns

**Limitations Weaknesses:**

1. It seems the proposed framework is only for language-based agent. Can it be generalized to digital (e.g., computer-use) or physical scenarios?

**Strengths Contributions:**

1. This paper is introduced to analyze a challenging problem in evaluating the generalization of cooperative agent in zero-shot or novel social environment.
2. It has decomposed generalization problems of multi-agent environment as substrate-aware and population-aware generalization. Multiple assumptions are introduced to guarantee the evaluation of multi-agent evaluation, including cooperation-eliciting substrates, cooperation-oriented populations, cooperation-eliciting scenarios and cooperative behavior scores.
3. A contest is developed to support the evaluation of cooperative intelligent for AI Agents, which involve focal agent design and scenario design. It also provide detailed experimental analysis from the contest, based on participant submissions. The results also demonstrate the limitations and potential of evaluating cooperative LLM-based agents.

---

> ### Author Rebuttal · Authors · 2025-07-30
>
> We thank Reviewer Sm24 for their review and for recognizing how our approach leverages cooperation-eliciting substrates, scenarios, and cooperation-oriented populations to assess zero-shot cooperation.
>
> Reviewer Sm24 correctly identifies that the proposed framework, in its current form, is exclusively for language-based agents. The Concordia architecture separates agent cognition from environment interaction through the Game Master (GM) abstraction, meaning extensions to new modalities require only implementing appropriate observation and action interfaces. For computer-use scenarios, the GM could provide screen state observations and translate agent intentions into UI interactions. For physical scenarios, observations could include sensor data while actions map to robotic controls. Essentially, there is nothing preventing this framework from expanding beyond LM agents and into digital or physical scenarios. The core evaluation principle of measuring generalization across contexts and populations transfer directly to these domains, and we hope that in future iterations of Concordia.
>
> Regarding the concern of track suitability, we want to clarify that the NeurIPS Competition explicitly recommends the Datasets and Benchmarks Track as the preferred medium for paper submission. Additionally, our research proposes an evaluation framework designed to benchmark the generalization and cooperative capabilities of LLM-based agents. This content aligns closely with the datasets and benchmarks content, meeting track expectations.

---

### Official Review · Reviewer_RddT · 2025-07-03

**Rating:** 6
**Confidence:** 4

**Summary:**

This paper presents the Concordia Contest, a benchmark for evaluating the generalization capabilities of LLM-based agents in mixed-motive cooperative scenarios. The authors argue that existing evaluation methods fail to measure how well LLM agents can cooperate in zero-shot settings with unfamiliar partners. They introduce a framework consisting of five text-based environments that test different aspects of cooperation. The paper reports empirical results from the NeurIPS 2024 Concordia Contest with 25 participating teams, revealing significant gaps in current agent capabilities, particularly in scenarios requiring persuasion and norm enforcement.

**Dataset Code Accessibility:**

NA; not applicable to this submission (e.g., no new dataset, benchmark, code, or data provided)

**Ethical Considerations:**

No, there are no or only very minor ethics concerns

**Final Justification:**

My questions are adequately answered. Thank the authors for organizing the contest beyond normal benchmarking.

**Limitations Weaknesses:**

1. The empirical experiments are solely using one LLM, Gemma2-9B, as the base LLM. Including more base LLMs could make the conclusions in this work more generalizable.

2. The authors note that "multiple participants reported that their agents scored markedly higher on the development phase scenarios than during the evaluation phase" in Section 7, suggesting potential overfitting.

**Strengths Contributions:**

1. The paper introduces a novel approach to evaluating cooperative generalization in LLM agents, extending concepts from the Melting Pot benchmark to natural language interactions.

2. The five substrates are well-motivated and cover diverse cooperation challenges.

3. This work is overall well-written with clear structure and informative tables/figures.

---

> ### Author Rebuttal · Authors · 2025-07-30
>
> We sincerely thank reviewer RddT for their thoughtful review and engagement in our work. We are particularly grateful with the acknowledgement of the set of substrates which were extremely carefully designed to cover a diverse suite of cooperative relevant situations for agents to navigate.
>
> We acknowledge that using only Gemma-2 represents a limitation, however, given the constraints of having to run the competition we had to select just one model. Gemma-2 was selected after careful evaluation of other SOTA models at the time of contest inception, including Llama and Mistral models, and emerged as the most performant model per cost. The Concordia contest is designed to assess the development of generalizable cooperative agents, and was not designed as a comparison of LM performance. The single model design choice serves critical purposes for fair evaluation: (1) it ensures a level playing field by preventing teams with larger computational budgets from gaining unfair advantages, (2) it provides consistency across all agent evaluations, eliminating model capability as a confounding variable, and (3) it makes the contest accessible to broader participation. We have updated our conclusion to make it clear that future work should run cross model comparison within the Concordia framework, as per your suggestion. We will also add the above clarifications on how Gemma was selected and why one model was used to run the contest to section 4. The 2024 NeurIPS Concordia Contest.
>
> Your observation about participants reporting higher development phase scores than evaluation phase scores provides valuable validation of our contest design rather than indicating a flaw. This pattern demonstrates that our development-evaluation split successfully identifies which agents achieve genuine cooperative generalization versus those that overfit to specific scenarios. Agents performing well only in development likely exploited regularities that did not generalize, while top-performing agents like taehun_cgcal demonstrated robust strategies that transferred across contexts. This finding reinforces the importance of our "veil of ignorance" principle and confirms we are measuring true cooperative intelligence rather than scenario-specific optimization.
>
> Thank you for your strong endorsement of our work. We sincerely appreciate your time and effort.

---

### Official Review · Reviewer_QqUo · 2025-07-03

**Rating:** 4
**Confidence:** 4

**Summary:**

The main idea of this paper is to evaluate the generalization capabilities of Large Language Model (LLM) agents in mixed-motive scenarios, especially when these agents interact with human and other artificial agents. The paper highlights that existing evaluation methods fail to adequately measure how well LLM agents' capabilities generalize to novel social situations.
To address this gap, the paper introduces the "Concordia Contest" as a principled evaluation method. This contest utilizes Concordia, a natural language multi-agent simulation environment, to test the ability of LLM-based agents to cooperate in zero-shot, mixed-motive environments.
The proposed method measures human-appropriate cooperative intelligence by testing an agent's ability to identify and exploit opportunities for mutual gain across diverse partners and contexts. Empirical results from the NeurIPS 2024 Concordia Contest show that while LLM agents achieved some success in tasks like negotiation, there are significant gaps in their robust generalization for reliable cooperation, particularly in complex coordination scenarios demanding persuasion and norm enforcement.

**Dataset Code Accessibility:**

Yes

**Ethical Considerations:**

No, there are no or only very minor ethics concerns

**Final Justification:**

This is a good paper for the community and I will keep my score.

**Limitations Weaknesses:**

- Explore how LLM agents can scale to operate effectively in larger group settings.
- Since the current Concordia environment relies solely on text, consider incorporating non-verbal cues and other multimodal inputs to make the social interactions more realistic and akin to human interaction. Can you explain more on this?
- Develop agents that are more adept at dynamic persuasion? which might effectively enforce social norms, and are capable of strategic coalition formation?

**Strengths Contributions:**

A principled argument for measuring human-appropriate cooperative intelligence, emphasizing the importance of generalization in multi-agent social interaction.

The introduction of a novel evaluation framework with five LLM-simulated environments, designed to test various aspects of cooperation such as strategic communication, social coordination under uncertainty, negotiation, and collective action, using a "veil of ignorance" approach for zero-shot generalization.

Detailed empirical evidence from the NeurIPS 2024 Concordia Contest, which reveals the strengths and limitations of current LLM agents and the framework.

---

> ### Author Rebuttal · Authors · 2025-07-30
>
> We thank reviewer QqUo for their review of our work and acknowledgement of our novel evaluation framework. We appreciate that you highlighted the design behind the "veil of ignorance" approach for zero-shot generalization, the variety of cooperation relevant properties that we cover, and our detailed empirical evidence that reveals current gaps in the capabilities of agents.
>
> We also thank reviewer QqUo for outlining specific limitations and opportunities for future development, and are excited to share how our framework can, with minor improvements, support these extensions. Regarding scaling to larger group settings, the Concordia architecture places no fundamental limits on agent numbers, as the Game Master (GM) can coordinate interactions among populations of any size. Current limitations to smaller groups stem from computational constraints rather than architectural restrictions. We are actively exploring techniques to enable larger-scale experiments that would reveal emergent phenomena like coalition formation and complex norm dynamics that arise only in larger populations.
>
> The suggestion of incorporating non-verbal cues and multimodal inputs represents an excellent recommendation. The fundamental architecture of Concordia makes these extensions straightforward to implement. Since the GM serves as the central mediator for all observations and actions in the environment, adding non-verbal communication channels requires only extending the GM's repertoire of observation types and action resolutions. The GM already translates between agent outputs and world states, so incorporating visual observations, gesture recognition, or facial expression interpretation involves implementing new observation interfaces while maintaining the same core evaluation framework.
>
> The observation regarding dynamic persuasion capabilities of agents is an area of extreme interest. As contest organizers, we had limited control over the specific strategies participants chose to implement, though we are pleased to report that several submitted agents demonstrated sophisticated approaches to these challenges. Notably, GEM_NegotiatorReputationFinal integrated negotiation techniques with memory tracking to dynamically enforce cooperative norms across interactions. The agent Synthetic_tom employed Theory of Mind reasoning to infer other agents' goals and beliefs, enabling more targeted persuasion strategies. Additionally, taehun_cgcal (the contest winner) successfully balanced coalition building with individual objectives by inferring common ground between agents. Details of the agents can be located in the appendix.

---

### Official Review · Reviewer_aqTA · 2025-07-05

**Rating:** 4
**Confidence:** 2

**Summary:**

This paper introduces Concordia, a natural language multi-agent simulation environment, to evaluate the generalization capabilities of LLM-based agents in mixed-motive scenarios. The NeurIPS 2024 Concordia Contest utilized this environment to test agents' ability to cooperate and achieve mutual gains across diverse situations like negotiation and collective action problems. Findings revealed that while some agents performed moderately well in negotiation, they struggled with complex coordination requiring persuasion and norm enforcement, highlighting gaps in current LLM agent capabilities for robust cooperative generalization.

**Dataset Code Accessibility:**

Yes

**Ethical Considerations:**

No, there are no or only very minor ethics concerns

**Final Justification:**

My concerns have been addressed. Based on the evaluation of early and rebuttal phase, I will maintain my original positive score.

**Limitations Weaknesses:**

1. The tags in Figure 2 are not adequately explained, making the figure difficult to interpret.

**Strengths Contributions:**

1. This paper studies the problem of generalization capabilities of LLM-based agents in mixed-motive social interactions, which is an interesting and emerging topic in the multi-agent systems and artificial intelligence domain
2. The paper is written with good clarity and thus it is easy to follow.
3. The experimental results generally demonstrate the effectiveness of the proposed contribution.

---

> ### Author Rebuttal · Authors · 2025-07-30
>
> We thank reviewer aqTA for their review of our work. We are excited by the recognition of our evaluation framework and the acknowledgement of the importance of this emerging area. We also appreciate your comment on the writing of the paper and how our empirical results demonstrate the effectiveness of our evaluation framework.
>
> We thank reviewer aqTA for the outlined limitation, and agree that tags in Figure 2 are not adequately explained. To address this, we have expanded Section 5. Evaluation Methodology and Results, to include a comprehensive explanation of our tagging system. Specifically, we now explain that during the design phase, each substrate was annotated with tags indicating the cooperative capabilities we hypothesized would be required for strong performance. These tags were 'pre-registered' since tagging was completed before the competition began, ensuring they were not influenced by participants' submissions or performance.We include the following language:
>
> “During the design phase, each substrate was annotated with tags indicating the (cooperative) capabilities we hypothesised would be required for strong performance. Tags were ‘pre-registered’, i.e., tagging was completed during task design, before the competition began, and thus was not influenced by participants’ submissions or their performance. These tags proved useful for interpreting performance patterns across agents: we observed in Figure 2 that agents struggled more on substrates that required persuasion or convention following compared to substrates that required negotiation.”
>
> We have also added an appendix item which includes an elaborated explanation of each individual tag. For example, "discouraging antisocial behavior" identifies scenarios where agents must recognize and appropriately sanction non-cooperative actions to maintain cooperative equilibria. The "persuasion" tag marks environments requiring agents to convince others through reasoned dialogue, testing their ability to construct compelling arguments while maintaining cooperative intent. The "calculation" tag indicates scenarios demanding quantitative reasoning about payoffs and strategic outcomes. We provide similar detailed explanations for all tags.

---

### Note · Authors · 2025-08-16

We thank the reviewers for their time and effort in assessing our work. We particularly appreciate the positive recognition of our framework and the validation of the importance of this emerging area. The reviewers’ acknowledgement of our work’s novelty and timeliness reinforces the value of establishing benchmarks for language-based agents.

We note that the primary points raised were requests for clarification rather than substantive concerns. We have prepared comprehensive responses to all points and will incorporate these improvements in the camera-ready version. Reviewer Sm24’s question regarding digital/physical extensions and Reviewer QqUo’s interest in larger group dynamics align closely with our development roadmap, validating our architectural choices and demonstrating engagement with the framework’s potential. The single-model limitation noted by multiple reviewers was a deliberate design choice to ensure fairness and consistency during the NeurIPS 2024 competition, which we have justified in our rebuttals. In the conclusion, we have also outlined opportunities for future multi-model studies. Importantly, no reviewer identified fundamental flaws in our methodology, dataset construction, or theoretical framework. We believe that the additions and clarifications provided address the most significant points raised during the review process.

---

### Decision · Program_Chairs · 2025-09-18

**Decision:**

Accept (poster)

**Comment:**

This paper introduces Concordia, a LLM MAS environment to evaluate the generalization capabilities of LLM-based agents in mixed-motive scenarios.
Authors argue that existing evaluation methods fail to measure how well LLM agents can cooperate in zero-shot settings with unfamiliar partners.
They introduce a framework consisting of five text-based environments that test different aspects of cooperation.
The paper reports empirical results from the NeurIPS 2024 Concordia Contest with 25 participating teams, revealing significant gaps in current agent capabilities, particularly in scenarios requiring persuasion and norm enforcement.


All reviewers agree that paper studies the problem of generalization capabilities of LLM-based agents in mixed-motive social interactions, which is an interesting and emerging topic in the multi-agent systems and artificial intelligence domain
Additionally, the paper was praised by the reviewers for several contributions:
First for the introduction of a novel evaluation framework with five LLM-simulated environments, designed to test various aspects of cooperation such as strategic communication, social coordination under uncertainty, negotiation, and collective action, and for using a "veil of ignorance" approach for zero-shot generalization. The five five LLM-simulated environments are complementary and test different aspects of agent cooperation.
Second, for the detailed empirical evidence from the NeurIPS 2024 Concordia Contest, which reveals the strengths and limitations of current LLM agents and the framework.
Also for the experimental results that generally demonstrate the effectiveness of the proposed contribution.

Reviewers identified few limitations, most of them were actual considerations for future work. Two main limitations were identified and discussed in the rebuttal:
a. The empirical experiments are solely using one LLM, Gemma2-9B, as the base LLM. Including more base LLMs could make the conclusions in this work more generalizable.
b. The current Concordia environment relies solely on text. Reviewer QqUo suggested considering incorporating non-verbal cues and other multimodal inputs to make the social interactions more realistic and akin to human interaction.
Further suggestions included to explore how LLM agents can scale to operate effectively in larger group settings, and develop agents that are more adept at dynamic persuasion (enforce social norms, strategic coalition formation).


Reviewers acknowledged authors' rebuttal, but none changed their scores.